# ALADDIN: JOINT PLACEMENT AND SCALING FOR SLO-AWARE LLM SERVING

## ABSTRACT

The demand for large language model (LLM) inference is gradually dominating artificial intelligence workloads, creating an urgent need for cost-efficient inference serving. While prior work focuses on single-worker optimization, it often overlooks cluster-level coordination across both queries and computing resources. Scheduling requests without considering their uncertainty can lead to SLO violations or overprovisioning, resulting in excessive cost.

In this paper, we present **Aladdin**, a scheduler that co-adaptively places inference queries and scales computing resources under probabilistic SLO constraints. Aladdin explicitly models request-level uncertainty through stage-wise latency distributions, and places queries based on their statistical profiles to maximize per-worker utilization. To improve robustness and cost-efficiency, we design a flexible constraint interface that supports distribution-aware tail modeling and risk-adjusted capacity allocation. Experiments show that Aladdin reduces serving cost by up to 71% under the same SLO level compared to standard baselines, which can translate to millions of dollars in annual savings.

## 1 INTRODUCTION

The rise of Large Language Models (LLMs) has rapidly transformed work and life OpenAI (2023), with their inference increasingly dominating AI workloads. Unlike conventional DNNs He et al. (2015), LLMs with billions of parameters require GPU memory and compute heavily, and GPU shortages have become common in both public and private clouds new york times (2023) consequently. Therefore, cost-efficient, scalable LLM serving becomes an urgent challenge. Recent works improve LLM inference efficiency by batching and scheduling. Continuous batching Yu et al. (2022); Kwon et al. (2023); Agrawal et al. (2023) improves GPU utilization but struggles with heterogeneous output lengths. FlexGen Sheng et al. (2023b) aggregates CPU and GPU resources to reduce cost. Split-phase systems Patel et al. (2023); Zhong et al. (2024); Hu et al. (2024) decouple prefill and decode, improving throughput, but rely on simple placement heuristics like JSQ or power-of-two Hu et al. (2024). These methods optimize throughput, with SLO improvement as a by-product. In parallel, classical SLO-aware serving Gujarati et al. (2020); Zhang et al. (2019); Romero et al. (2021); Crankshaw et al. (2017) assumes predictable workloads, while cluster-level scheduling work Grandl et al. (2014); Jyothi et al. (2016) focuses on traditional jobs with fixed resource profiles. These frameworks are inadequate for the heterogeneous, memory-intensive, and delay-sensitive nature of LLM inference. **Key challenges** remain: **(1)** decoding KV cache can easily overflow under poor request placement; **(2)** decoding latency increases with token count even at fixed batch sizes; **(3)** optimal worker configuration must account for both compute and memory trade-offs. Existing approaches largely neglect these interactions.

**This paper presents ALADDIN**, a co-adaptive LLM serving system that jointly performs probabilistic SLO-aware request placement and resource scaling. Aladdin models latency distributions at a fine-grained stage level and leverages multi-dimensional constraints (e.g., KV cache, ATGT) to minimize GPU cost. Unlike prior work, it guarantees SLO satisfaction across all queries and adapts to dynamic workloads with provable efficiency. As shown in Figure 1, when LLM inference requests arrive, Aladdin first predicts minimal computing resources by learning the optimal configuration of serving workers based on the historical input-output length distributions and the request arriving rate. Secondly, Based on the requests' input and predicted output length, as well as the learned

batching performance models, we formulate the request placement to an online multi-dimensional bin packing problem. Lastly, We monitor the ongoing requests of each worker and adjust the placement of new arrivals to reduce the impact of output length prediction errors. Aladdin supports the default setting vLLM Kwon et al. (2023) that does the prefill and decode in the same worker, as well as the decoupled prefill and decode setting like Patel et al. (2023); Zhong et al. (2024); Hu et al. (2024).

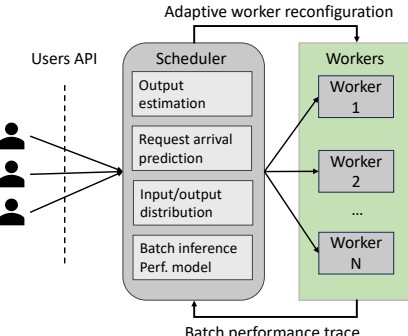

Figure 1: The overall architecture of co-adaptive scheduling

Overall, the main contributions of our paper are:

- We conduct an empirical study of the dynamic batching performance of prefill-decoding LLM inference and deduce the accurate performance prediction model of LLM serving.

- We design a near-optimal online algorithm and a novel scheduler, Aladdin, to co-adaptively place the queries and manage computing resources to fulfill all requests' SLOs using minimal GPUs.

- We conducted a comprehensive evaluation of Aladdin, including the validation of our LLM inference performance models on the A100 and V100 testbeds to establish its generality. We evaluated Aladdin's end-to-end performance with the real-world workload, which arrived as a stream on GPU servers. Additionally, we conducted a large-scale simulation for the high-demand LLM serving scenario.

## 2 BACKGROUND AND MOTIVATION

**LLM Inference SLOs:** In contrast to other DNN inference workloads Gujarati et al. (2020) that have well-defined latency targets, LLM inference is a two-stage iterative process. The first stage involves the generation of the initial token, which processes all prefiled tokens, while the second stage is the decode stage, where tokens are generated iteratively one by one. LLM inference latency depends on the output length. Although the time for generating the first token increases with the number of prefiled tokens Agrawal et al. (2023), it remains predictable based on the length of the prefiled tokens. Additionally, the first token generation is a single-round inference process without iteration, so we have set a predetermined response deadline for time to the first token (TTFT). For the decoding process, previous work Patel et al. (2023) adopts the time between tokens (TBT) metric, constraining the latency between every token smaller than the target. However, the TBT metric is an over-strict metric with less flexibility, and it does not directly affect the user's quality of experience. We introduce the quality of experience SLO using the average token generation time (ATGT) metric $ATGT = \frac{t_{decode}}{l_{out}-1}$, where $t_{decode}$ is the decode time of a request and $l_{out} - 1$ is the output length of the decode phase. This metric reflects the average time spent generating each token during the decode stage. For example, the average reading speed for individuals is approximately four words per second Brysbaert (2019). To ensure the delivery of quality service, the average token generation time for each request must not exceed 0.2 seconds. **Output Length Prediction:** The input and output lengths of requests have a huge impact on the decision of the inference requests and worker configuration. However, when we make the request placement decisions, we only have

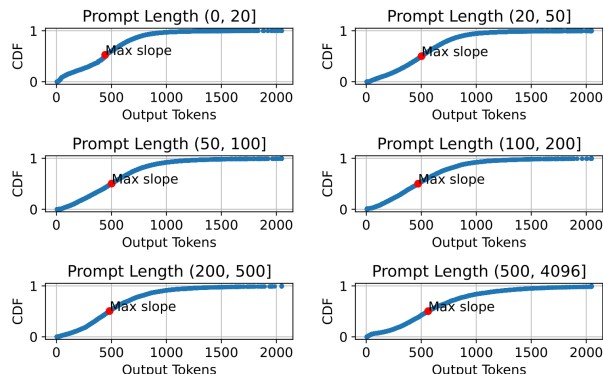

Figure 2: CDF of output length for different prompt Lengths from ShareGPT and llama2-13b-chat-hf generated output.

the information for the input length of each request. There are some techniques to predict the output length of each request. Previous work Zheng et al. (2023); Hu et al. (2024); Qiu et al. (2024) proposed the response length perception that harnesses the output length prediction before the execution ability of LLMs. They use historical data to fine-tune the LLM. However, there are drawbacks to this methodology. Firstly, the overhead of using a LLM to predict the output length is non-negligible because the output length prediction process is another inference. Although previous work Hu et al. (2024) uses a smaller model to predict the output length for a larger LLM, the prediction overhead is still significant. And the prediction of response length perception is out of control. From our experiment result, the response length predicted by the fine-tuned models is biased. Figure 2 presents the CDF of output length given the corresponding prompt length in different ranges. Although the output length prediction error is inevitable in our request placement, the prediction without bias can partially cancel the prediction error when we put requests in a batch. Hence, we use the estimated output length of each input length in the historical data as the predicted output length. This is the most naive output length predictor. Although the prediction error may be high, this prediction method has a low overhead and is non-biased. In Section D, we address the prediction error by designing a novel re-balancing algorithm. Note that the output length prediction is not the main contribution of this paper. If there are accurate, non-biased, and low overhead output length predictors in the future, the performance of Aladdin could be further improved.

## 3 CONTINUOUS BATCHING PERFORMANCE MODELING

In LLM inference, The transformer uses the given prompt (context) as the initial input and generates additional tokens one by one. During the inference process, the transformer performs self-attention, which requires the key-value (KV) vectors for each token (prompt and generated tokens) in the current sequence. These vectors are stored in the GPU as two matrices (key matrix and value matrix) during inference, often called the KV cache. At the beginning of an inference, the KV cache stores the key and value matrices of the prompt tokens. During response generation, the KV vectors associated with that token are appended to the KV cache matrices with each token generated. This dynamic expansion leads to a linear relationship between the KV cache's usage and the current sequence size. This linear relationship signifies that the KV cache's memory footprint increases proportionally with the sequence length. So the KV cache usage of a request

$$kv = h(l_{in} + l_{out}) + j, \tag{1}$$

where $h$ and $j$ are learnable coefficients, and $r$ is the output tokens generated so far.

Iteration-level batching poses unique challenges. Not all requests can be batched together at any iteration due to varying input shapes. Orca Yu et al. (2022) addresses this by proposing selective batching. However, operators like Attention require inputs with identical shapes, leading to separate calculations using cuBLAS NVIDIA (2023) routines for batch matrix multiplication. The separate multiplications for each request result in a linear scaling of iteration time to the batch size. In default

settings like vLLM Kwon et al. (2023) or split-phase inference, one batch can only contain prefill or decode. Since the query in the attention mechanism of the prefill process is a matrix that includes all input tokens, the query of the decode process is a vector of the last generated token. The iteration latency model of the prefill and decode batch is different.

**Prefill iteration time.** Since prompt processing is a computing-bottleneck process, a single request with a reasonable input length can effectively saturate the worker's computing power, which means the batching effect has limited improvement to the throughput in the prefill process. Our preliminary results indicate that the iteration time of the prefill batch is not affected by the batch size and is linear with the total input length of all batched requests. The iteration time:

$$t_{pre} = k_1 \sum l_{in} + c_1, \tag{2}$$

where the $\sum l_i n$ is the total input length of all requests in the prefill batch, $k_1$ and $c_1$ are the learnable coefficients.

**Decode iteration time.** However, the token generation process has low compute utilization since each query only generates one token in an iteration. With a fixed batch size, the iteration time linearly increases as the average context length (the input length of the request and the tokens generated so far) increases. Similarly, with the same average context length, the iteration time increases linearly with the batch size. According to the experiment, the iteration time with a batch size of one (i.e., single request inference without batching) remains nearly constant. With this information, when we haven't reached the KV cache limit, the iteration time $t_d$ is:

$$t_d = (k_2 l_{ave} + c_2)b + c_3, \ b > 1, \tag{3}$$

where $b$ is the batch size, $l_{ave}$ is the average context length among all requests. $k$ and $c$ are learnable coefficients. In the scheduling algorithm design, given the ATGT SLO $T_{dec}$, the total input length is limited by a function of batch size $b$:

$$l_d \leq \frac{1}{k_2} \left( -c_2 b + T_{dec} - c_3 \right), \ b > 1. \tag{4}$$

Note that all coefficients in Eq. 4 are positive according to the batch inference scaling. And $T_{dec}$ must be greater than $c_3$ because the decoding latency SLO we choose must be greater than the individual request decoding latency without batching. From Eq. 4, we deduce that with a larger batch size, the maximum total input length limit of all requests within the batch decreases.

## 4 CO-ADAPTIVE SCHEDULING

When requests arrive at the scheduler, our task is to determine how to use the minimum number of GPUs to serve both newly submitted and ongoing requests while ensuring compliance with the SLO requirements. This overarching objective can be deconstructed into several critical components: (1) We need to determine the minimal GPU number required to serve the queries that fulfill the SLO requirements. (2) Find the most efficient configuration of these GPUs, such as the number of workers and the number of GPUs configured with each worker. (3) Decide how to place the requests to each worker in a manner that optimizes the utilization of each worker.

It's important to note that these three components are interconnected. When one decision is made, the other two are simultaneously determined. For example, when we establish the total number of GPUs, this decision implicitly dictates the optimized placement of GPUs and models on each worker, as well as the optimization of request assignments to each worker. Conversely, if we can devise a more effective strategy for worker configuration or request assignment that enhances resource utilization, we can reduce the total resource requirements for a more cost-efficient service. Firstly, Let's look into the optimal single-worker configuration because the optimal configuration for each worker is orthogonal to the request scheduling and worker number determination.

### 4.1 WORKER CONFIGURATION

In this paper, we consider the tensor parallelism distributed inference. The optimal worker configuration is achieved at the optimal per-GPU throughput. With the different ranks of tensor parallelism, the computing, communication, and KV cache capacity all impact the throughput. In the default

vLLM Kwon et al. (2023) setting, the prefill and decode processes share the same worker, but decode dominates since it generates tokens iteratively while prefill runs only once. We have to predict the parallelism strategy with the most per-GPU throughput for decode phase. In tensor parallelism, each GPU computes its split tensor locally, then aggregates results via All-reduce. The compute time scales inversely with the number of GPUs:

$$t_{compute} = \frac{k_4}{N_g} + c_4, \tag{5}$$

where $N_g$ is the number of GPUs per worker. The All-reduce communication overhead is $(N_g - 1)/N_g$; while nearly constant for large $N_g$, it is non-negligible on modern servers (e.g., DGX A100/H100) with at most 8 GPUs, though intra-node bandwidth mitigates straggler effects. The KV cache capacity is $M = N_g m_{gpu} - m_{model}$. Throughput may be limited either by KV cache or by the iteration SLO: in the former case when KV cache is full, in the latter when the decode iteration time reaches the ATGT latency limit.

The maximum per-GPU throughput of tensor parallelism rank $N$ is:

$$T_{max} = min \left\{ \frac{M}{N_g m_r(t_{compute} + t_{comm})}, \frac{B}{N_g T_{decode}} \right\}, \tag{6}$$

where $m_r$ is the average per request KV cache demand learned from the historical data, and $t_{compute} + t_{comm}$ is the iteration time given the batch size $\frac{M}{m_r}$ with $N_g$ GPU per worker. $T_{decode}$ is the ATGT SLO, and $B$ is the batch size corresponding to the SLO. The optimal worker configuration has $N_g^{opt}$ GPUs that maximize $T_{max}$. Note that with homogeneous GPUs, the optimal worker configuration is independent of request arrival rate but depends on model size, context length, and GPU compute and memory capacity. Thus, when adapting to varying workloads, each worker's configuration remains fixed.

## 4.2 REQUEST PLACEMENT POLICIES

We optimized the worker configuration to achieve maximum per-GPU throughput, and our next objective is to minimize the number of workers required for LLM service. The placement of queries to workers significantly affects efficiency of resource utilization. Figure 4 illustrates the suboptimal of naive JSQ and reveals the optimal request placement strategy. In this example, requests need to be placed to two workers with KV cache capacity of 9. If we adopt JSQ, two long prompt requests will be placed to the same worker, while two long output requests will be placed to another worker. Suppose a token requires 1 KV cache capacity. The max KV cache demand for both workers is 10 when requests finish generation, which exceeds the KV cache capacity of 9. Therefore, we need to move requests to the waiting queue until there is available KV cache. However, with the optimal request placement, a long prompt request and a long output request are placed in one worker. The max KV cache demand for each worker is 7. We leverage the parameters notated in Table 2 in Appendix B and the following information: (1) Learnable prefill time to total input tokens Eq. 2, input tokens limit to batch size when constraining the decode iteration time Eq.4 and learnable KV cache usage to token count Eq.1 functions for each group. (2) The current KV cache usage $m = \sum kv$ and total KV cache $M$ for each worker. (3) For each newly added request, we utilize the known input prefill length $l_j^{in}$ and predicted output length $l_j^{pred}$. For ongoing requests, we take into account the current length generated $l_j^{out}$.

The request scheduling with the constraints can be seen as a multi-dimensional bin packing problem. We formulate it as a mixed integer programming (MIP) that schedules the new-arrived requests between the scheduling heartbeat with different input/output lengths $l_{in}$ and $l_{out}^{pre}$, and we want to minimize worker number $W$. Let $x_{ij}$ be a binary variable that equals 1 if request $j$ is scheduled to Worker $i$, and 0 otherwise. Let $y_i$ be a binary variable that equals 1 if Worker $i$ is used, and 0 otherwise. Assume $I$ is the initial worker number larger than the optimal $W$. When there are ongoing requests, for an ongoing request $j$, to prevent the unnecessary migration between workers, the $x_{ij}$ is kept the same as the current decoding worker. We also need to guarantee that the new request's prompt processing time won't violate the token generation time SLO. The MIP problem

can be formulated as follows:

$$\min \quad \sum_{i=1}^{I} y_i$$

$$\text{s.t.} \quad \sum_{i=1}^{I} x_{ij} = 1, j = 1, 2, \ldots, J, \quad \text{(a)}$$

$$\sum_{j=1}^{J} x_{ij}(l_j^{in} + \gamma l_j^{out}) \leq \theta l^d \left( \sum_{j=1}^{J} x_{ij} \right), i = 1, 2, \ldots, I, \quad \text{(b)}$$

$$t_p \left( \sum_{j=1}^{J_{new}} x_{ij} l_j^{in} \right) \leq T_{pre}, i = 1, 2, \ldots, I, \quad \text{(c)}$$

$$t_p \left( \sum_{j=1}^{J_{new}} x_{ij} l_j^{in} \right) \leq \theta min(T_{dec} l_{ij}^{out} - t_{ij}^{dec}), i = 1, \ldots, I, \quad \text{(d)}$$

$$\left[ \sum_{j=1}^{J} \mathbf{w}_j x_{ij} \right]_k \leq M, k = 1, 2, \ldots, K, i = 1, 2, \ldots, I, \quad \text{(e)}$$

$$x_{ij} \leq y_i, i = 1, 2, \ldots, I, j = 1, 2, \ldots, J, \quad \text{(f)}$$

$$x_{ij} \in \{0, 1\}, i = 1, 2, \ldots, I, j = 1, 2, \ldots, J, \quad \text{(g)}$$

$$y_i \in \{0, 1\}, i = 1, 2, \ldots I. \quad \text{(h)}$$

The constraints are: (a) Each request must be scheduled to one worker. (b) According to Eq. 3, the iteration time is determined by both batch size and the total context length. Eq. 4 shows the maximum total context length of all requests in one batch given the batch sizes. This constraint ensures the ATGT SLO for the decode process. Since the iteration time increases as more tokens are generated during decoding, the coefficient $\gamma$ can be considered as a "strictness knob" that tunes the scheduling bound, $0 \leq \gamma \leq 1$. When $\gamma = 0$, only the first iteration can meet the ATGT SLO. When $\gamma = 1$, the last token generation time can meet the ATGT SLO. We normally set $\gamma = 0.5$ to increase the worker utilization while guaranteeing the SLOs. (c) According to Eq. 2, the sum of all new requests' input is limited by the TTFT SLO. (d) Since the prefill of new requests preempts the decode for ongoing requests, the prefill time of new requests can not exceed the time that ongoing requests have saved compared with the ATGT limit. Reflecting on the limitation of the sum of new requests' input length. (e) The total KV cache demand of all the requests scheduled to each worker cannot exceed the KV cache capacity $M$. $K$ is the sequence length limit of the serving model. $\mathbf{w}$ is the vector with length $K$ that shows a request's KV cache footprint. For example, for request $j$,

$$\mathbf{w} = \begin{bmatrix} kv(l_j^{in}) & kv(l_j^{in} + 1) & \cdots & kv(l_j^{in} + l_j^{pred}) & 0 & \cdots & 0 \end{bmatrix},$$

where each element in the vector presents the KV cache demand of an iteration. The KV cache demand for the first iteration includes the KV cache for input tokens. The KV cache demand increases in the following iterations while output tokens are generated. The KV cache demand becomes zero when the request $j$ finishes. This constraint guarantees that for all scheduled iterations, the KV cache demand will not exceed the KV cache capacity of the worker. (f) If a worker is used, it should have at least one request scheduled. Otherwise, we don't need this worker. (g)(h) All variables are binary. Unused boxes will have $y_i = 0$ and will not be counted in the objective function. $0 < \theta < 1$ in (b)(d) is another hyperparameter that adapts to the prediction error of output length. For example, when $\theta$ is small, the constraints are tighter, so requests are less likely to violate the SLOs. However, the drawback is that we need more workers for the serving.

**Scheduling heuristic.** The multi-dimensional bin-packing problem is NP-hard, so an efficient heuristic is needed to approach optimal scheduling. Given that requests arrive in an online pattern, we employ the best-fit algorithm for online bin packing Letchford (2002). It schedules each arrived request to the worker with the maximum load and can guarantee the satisfaction of all SLO

---

**Algorithm 1:** Request scheduling heuristic

---

1 **Input:** $l^{in}$, $l^{pred}$ of the new request $j$. $l_{in}, l_{pred}, l_{out}$ of all ongoing requests. KV cache
capacity $M$ for each worker. Worker number $W$. Performance models $kv(t), t_{iter}(b, l), t_{pre}(l)$.
2 **Output:** Worker $i$ where job $j$ be scheduled, $x_{ij} = 1$.
3 **Initial:** $workerfound \leftarrow False$
4 Sort all bins on *capacity_norm* from large to small.
5 **for** *sorted bins* $i = 1, 2, \ldots, I$ **do**
6      **initial** $x_{ij} \leftarrow 0, i = 1, 2, \ldots, I$
7      $x_{ij}{=}1$
8      **if** ⓑ *and* ⓒ *and* ⓓ *and* ⓔ *for* $i$ **then**
9          $workerfound \leftarrow True$
10          **return** $x_{ij}$

11 **if** $workerfound = False$ **then**
12      Open a new bin $(I + 1)$ and add job $j$.
13      $workerfound \leftarrow True$
14      **return** $x_{(I+1)j} = 1$

---

constraints. Intuitively, this heuristic increases the utilization of each worker compared to other scheduling algorithms, such as joining the shortest queue, thereby reducing the number of workers.

In the multi-dimensional bin packing problem, determining the metric for each worker's load is non-trivial. Using the batch size of each worker as the metric for its load is sub-optimal because the input and output lengths of requests significantly influence each worker's load. We propose *capacity_norm*, which is the L2 norm of batch size $B$ and weighted context length $\sum (l_{in} + \gamma l_{out})$ of all ongoing requests to rank all workers. The heuristic algorithm for scheduling an arriving request is described in Algorithm 1.

To mitigate output length prediction errors, we design an error-aware rebalancing strategy that monitors worker over-/under-utilization and dynamically redistributes requests. Details of error metrics and the rebalancing algorithm are provided in Appendix D. Implementation details are in Appendix E. For further details on Aladdin's system design, including workflows for both continuous and split-phase inference, please refer to Appendix F.

### 4.3 TAIL LATENCY REMARKS

The heuristics above admit a chance-constrained interpretation. We model per-request end-to-end latency (TTFT/ATGT) as the sum of prefill, decode, and queueing stages, and enforce worker-level tail guarantees. For worker $j$ with placement vector $\mathbf{a}_j$, we consider a scheduling window with $n$ co-scheduled requests and let $\mathbf{X} \in \mathbb{R}^n$ be the random vector of their per-request latencies. Define $S_j := \mathbf{a}_j^\top \mathbf{X}$, $\mu_j := \mathbb{E}[S_j]$, and let $\Sigma := \mathrm{Cov}(\mathbf{X})$. We allocate a global violation budget $\delta \in (0, 1)$ across workers via nonnegative weights $\{\beta_j\}$ with $\sum_j \beta_j \leq 1$, and set $\tau_j := \beta_j \delta$. The tail factor $\gamma(\cdot)$ maps a target tail probability to a one-sided bound like Cantelli or a normal quantile:

$$\Pr\Big(S_j \geq \mu_j + \gamma(\tau_j)\sqrt{\mathbf{a}_j^\top \Sigma \, \mathbf{a}_j}\Big) \leq \tau_j.$$

This template subsumes truncated-moment modeling for long outputs, covariance-aware bounds for correlated requests, and dynamic risk allocation across workers. Instantiating $S_j$ for TTFT or ATGT ties our engineering knobs of batch sizing or worker configuration to explicit tail-risk controls.

Under mild conditions like non-degenerate variability and calibrated $\gamma$, the scheduler dominates naive point estimate baselines since it never worsens the number of active workers or the SLO attainment rate and is often strictly better. This bridges Sections 4.1, 4.2 with a tail-aware foundation. Order statistic and CVaR refinements are deferred to the appendix.

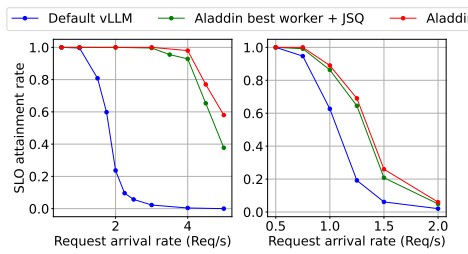 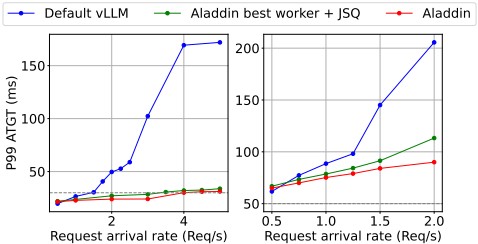

(a) The end-to-end SLO attainment rate, (left): LlaMa2-13b, (right): LlaMa2-70b

(b) The end-to-end P99 ATGT, (left): LlaMa2-13b, (right): LlaMa2-70b

Figure 3: End to end experiments on A100 testbed

## 5 EVALUATION

Our experimental setup details are provided in Appendix H. For the evaluation of Aladdin, we validate the accuracy of our performance modeling for continuous batching inference in Appendix I. Here, we examine the performance improvement achieved with Aladdin with different scenarios in Section 5.1 and Appendix J due to page limit. We also provide the overhead analysis of Aladdin in Appendix K. The primary information of our evaluation is as follows: (1) Aladdin accurately predicts performance metrics with the maximum error less than 10%. (2) Aladdin reduces the GPU number required by up to 71% and 60% compared with vanilla vLLM Kwon et al. (2023), and split-phase inference engines Zhong et al. (2024); Patel et al. (2023)'s decode instances for the same workload. (3) Although single-worker optimization techniques like chunked prefill Agrawal et al. (2023) and split-phase inference Patel et al. (2023); Zhong et al. (2024) reduce the cost for inference, the cost reduced by Aladdin is orthogonal to those techniques. Aladdin can be combined with single-worker optimization techniques to improve the performance further.

Moreover, we have another three experiments, details in appendix, working on (1) different language model other than LlaMa2, (2) comparison to the case with oracle output length prediction, and (3) ablation of rebalancing.

### 5.1 END-TO-END PERFORMANCE

We evaluate Aladdin's end-to-end performance by comparing it with baselines on our A100 and V100 testbeds. In this experiment, requests arrived on Aladdin in a stream format following Poisson distribution. We use ShareGPTteams (2023) dataset for the conversation content. The baseline we select is the default vLLM, with all GPUs (4 GPUs) on each machine in one worker. Since the performance improvement achieved by Aladdin is gained both from request placement and optimal worker configuration, we configure vLLM with the optimal worker configuration and adopt JSQ for the request placement to do the ablation study. Table 1 reveals the best worker configuration for different models on different testbeds.

Table 1: Optimal worker configuration for different models and GPUs for ShareGPT dataset

| Model | A100 (GPUs/worker) | V100 (GPUs/worker) |
|---|---|---|
| Llama2-70b-chat-hf | 2 | N/A |
| Llama2-13b-chat-hf | 1 | 2 |
| Llama2-7b-chat-hf | 1 | 1 |

The results of A100 testbed are shown in Figure 3. For the LlaMa2-70b model, Aladdin reduces the SLO violation rate by up to $3.5X$ compared with the default vLLM setting. Compared with the best worker configuration with JSQ placement, Aladdin only improved the SLO attainment rate by up to 19%. This is because there are totally two workers for the LlaMa2-70b model, which limits the improvement in the SLO attainment rate. However, Aladdin significantly reduces the P99 ATGT by up to 40% compared with JSQ, as shown in Figure 3b's right side. The results for the LlaMa2-13b

model are distinct from the 70b model. The optimal worker configuration for the 13b on the A100 testbed is one GPU according to Table 1. There are four workers in total for the request placement. So Aladdin improves the SLO attainment rate by up to $51\%$ compared with JSQ, but only has minor P99 ATGT improvement. The results of the V100 testbed are described in Figure 9. The difference is when the request arrival rate is low, the P99 ATGT of baseline default vLLM output performs the performance with optimal worker configuration. This is because when the arrival rate is low, the batch effect is not significant, and the worker with more GPUs has higher computing power than the worker with fewer GPUs. Nevertheless, in those arrival rates, both baselines and Aladdin fulfill all requests SLOs. The higher ATGT won't further improve the SLO attainment rate. Note that we don't include the P99 TTFT because vLLM Kwon et al. (2023) preempts the decode batch with the prefill batch when new requests arrive, making the ATGT more easily violate the SLO.

## 5.2 SCOPES AND EXTENSIONS

This work evaluates the scheduling framework in a controlled single-model, single-node setting. The formulation itself generalizes: each model induces its own stagewise latency and KV footprint, and the same chance-constrained placement applies once these profiles are specified. Extending the implementation to multi-model and multi-tenant deployments, including migration and fairness mechanisms, is therefore a natural next step when the required system support is available.

The prototype currently runs on a single node. Multi-node and cross-region deployments introduce interconnect and remote-KV effects that can be integrated into the same stagewise model, and validating them will require larger clusters or cloud environments. Modern runtime mechanisms such as paged attention, chunked prefilling, and KV streaming change slope parameters while preserving structural constraints; integrating these implementations and evaluating their impact is a natural extension toward production systems.

Finally, production traces often exhibit richer nonstationarity and long-context behavior. The probabilistic formulation already supports such variation through online residual tracking and risk budgeting, but a broader evaluation on production-like traces and the development of a full closed-loop controller for maintaining a target tail-violation rate represent promising directions for future work.

## 6 RELATED WORK

Recent LLM inference studies span three directions: performance modeling, SLO specification, and serving systems. **Performance modeling.** Prior work Narayanan et al. (2023) estimates prefill/decode latency by FLOPs-based modeling for single queries, ignoring hardware and multi-query interactions. DistServe Zhong et al. (2024) models batched latency under specific configurations, but lacks input/output length generality. Our work extends modeling to dynamic batches with heterogeneous lengths, enabling more general scheduling. **SLO specification.** Existing systems Patel et al. (2023); Zhong et al. (2024); Hu et al. (2024); Agrawal et al. (2024) adopt fixed TTFT or token-level latency (TBT) targets, but the latter is overly strict and misaligned with perceived QoE Sheng et al. (2023a). We instead propose ATGT as a more flexible and user-aligned SLO. **Serving systems.** Application-level optimizations (e.g., continuous batching Yu et al. (2022), page attention Kwon et al. (2023); Strati et al. (2024), chunked-prefill Agrawal et al. (2023; 2024)) focus on single-worker efficiency. Worker-level approaches Patel et al. (2023); Zhong et al. (2024); Hu et al. (2024); Oh et al. (2024) optimize intra-node GPU usage. Some works address query scheduling Liu et al. (2024); Qiu et al. (2024), but without co-adaptive placement and resource scaling. Our work unifies these aspects under a cost-aware, SLO-guaranteed framework.

## 7 CONCLUSION

We propose Aladdin, an adaptive LLM serving system that effectively scale and configures computing resources and optimally places inference queries to minimize serving costs while fulfilling SLOs. In this paper, we first deduce the performance models of the batched prefill and decode phases in LLM inference. Then, we predict the minimal computing resources required along with the corresponding worker configuration and request allocation. Results show that Aladdin reduced LLM serving costs by up to $71\%$ compared to state-of-the-art baselines.

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

## A  ADDITIONAL BACKGROUNDS

**Batch Processing of LLM Requests:**  The demand for large language model (LLM) serving has experienced exponential growth, making the efficient serving of LLM requests a critical challenge. LLM serving places significant demands on GPU computing power and memory, which can be prohibitively expensive. Previous work, such as Orca Yu et al. (2022) and vLLM Kwon et al. (2023), have introduced dynamic continuous batching techniques for transformer-based generative models to optimize GPU utilization. LLM generates responses iteratively, producing one token at a time and using it as input for the next iteration. Importantly, these requests may have varying output lengths, necessitating different numbers of iterations to complete. Traditional request-level batching methods pose a disadvantage. Requests within the same batch must wait until all requests are finished before results are returned. In contrast, continuous batching employs iteration-level scheduling, submitting an iteration calculation to the execution engine with each token generation. This approach prevents early-finish requests from waiting for the completion of other requests, improving GPU utilization.

There are challenges to improving the request placement and worker scaling.
**Challenge 1: Heterogeneous phases of LLM inference.** The transformer-based LLM inference consists of prefilling and decoding stages. The prefill stage is the first iteration of an inference request that processes all prompt tokens; it has more computing demand than the decoding process. The decoding process is a memory-intensive stage compared with the prefill stage because of the KV cache. These distinct features result in different performance models of prefilling and decoding processes for each request. Given the requests with various input and output lengths, accurately predicting the iteration time of batched prefill and decode is challenging.
**Challenge 2: Worker performance prediction.** The inference workload varies over time with high uncertainty. Meanwhile, worker configuration and the number of workers directly affect the cost of inference. Considering the request arrival pattern, we must take into account the worker's computing latency, KV cache capacity, and communication overhead. The search space for configurations is too large to be explored by a naive enumeration approach. Accurately predicting optimal configurations poses significant challenges.
**Challenge 3: Handle the error of output length prediction.** The output length prediction error is inevitable. Therefore, reducing the impact of prediction errors on output length is crucial for enhancing performance when assigning tasks to workers. Systems need to effectively react when the prediction error is detected.

## B  NOTATIONS

Table 2: The inputs to Aladdin and decisions Aladdin makes

| Inputs | Notation | Definition |
|---|---|---|
| | $kv(t)$ | The KV cache usage to tokens function |
| | $l_d(b)$ | The input length limit to batch sizes |
| | $t_p(l)$ | The prefill iteration time function |
| | $m_i$ | The KV cache usage of Worker $i$, $i \in W$ |
| | $M$ | The KV cache capacity of each worker |
| | $l_j^{in}$ | The input length of a request |
| | $l_j^{pred}$ | The predicted output length of a request |
| | $l_j^{real}$ | The real output length of a request |
| | $l_j^{out}$ | The output tokens a request generated so far |
| | $t_j^{dec}$ | The time spent for decoding phase so far |
| | $T_{pre}$ | The SLO of prefill latency |
| Outputs | Notation | Definition |
| | $W$ | The total worker number |
| | $x_{ij}$ | binary variable for request $j$ |
| | $y_i$ | binary variable for Worker $i$ |

## C   JSQ FOR REQUEST PLACEMENT

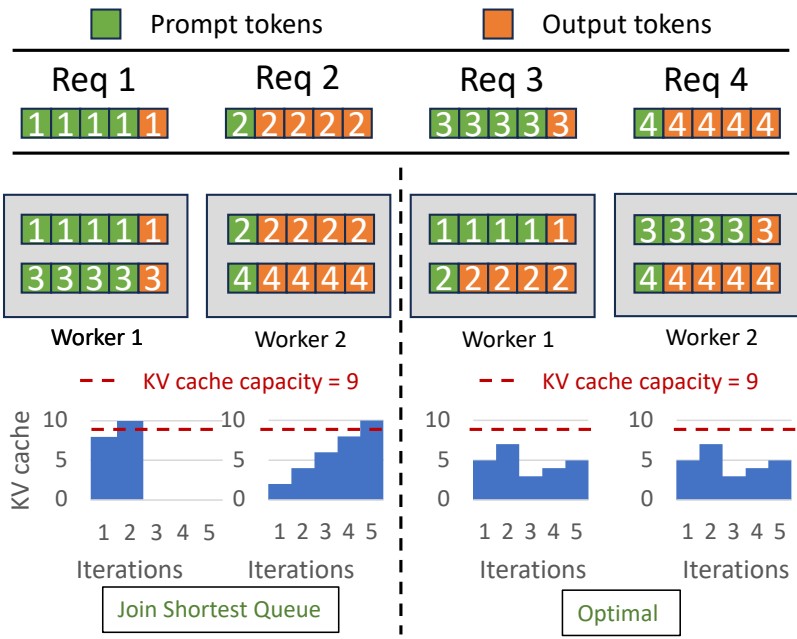

Figure 4: An example illustrates the sub-optimal of JSQ for request placement.

## D   ADDRESSING PREDICTION ERRORS

The output length cannot be accurately predicted before execution. If we overestimate the output length, worker utilization will be reduced. Conversely, there will be SLO violations. When an ongoing request in a batch finishes earlier than predicted, we mark this worker as overestimated. If an ongoing request's output length is underestimated, i.e., the request hasn't finished with the predicted tokens, we mark this worker as underestimated and predict the output length again. Before the execution of the new requests, we re-schedule new requests that have been scheduled to the over-utilized workers to the under-utilized workers. We use $l^e$ and $b^e$ as the metrics to indicate the estimation error of each worker, where $l^e$ is the accumulated error of output length for outstanding requests, and $b^e$ is the error of batch size for each worker. If Request $j$ is finished before the estimated iteration, which means we overestimate the output length, we can calculate the output length over-estimate error $l_j^{real} - l_j^{pred}$. If we underestimate the output length of Request $j$, we predict the output length $l_j'^{pred}$ again using conditional average output length when $l_j^{real} > l_j^{pred}$ with the same input length $l_j^{in}$. In the request scheduling, we use $l^e$ and $b^e$ as the indicators to balance the workload between workers and reduce the effect of output length prediction error. The calculation for $l^e$, $b^e$, and the re-balancing algorithm are described in Algorithm 2

## E   IMPLEMENTATION

Aladdin is specifically designed for single-model serving, eliminating any model cold start problem for each worker. We adopt vLLM Kwon et al. (2023) for dynamic batch inference to optimize the KV cache usage of each worker and make the KV cache usage more predictable. Aladdin's request scheduler is a scheduling layer on top of the vLLM inference engine. Users submit their requests to the Aladdin frontend through the API interface. Aladdin routes and schedules the requests to different workers through each server's API interface. Note that Aladdin is a non-blocking system; once a request is scheduled to a worker, it will start inference in the next iteration. Aladdin doesn't support request migration, which means once a request has been sent to a worker, we won't migrate it to another worker with the same duty.

---

**Algorithm 2:** Re-balancing with prediction error

1  **Input:** $x_{ij}, l_j^{pred}, l_j^{out}, l_j^{real}$ of $J_{old}$ ongoing requests. $x_{ij}, l_j^{in}, l_j^{pred}$ of $J_{new}$ new requests.

2  **Output:** Updated $x_{ij}$ of new requests.

3  **Initial:** $l_i^e = b_i^e = 0, i = 1, 2, \ldots, I.$

4  **for** *worker* $i = 1, 2, \ldots, I$ **do**

5     **for** *ongoing job* $j = 1, 2, \ldots, J_i$ *on worker* $i$ **do**

6         /*Check if under estimate output length*/

7         **if** $l_j^{out} > l_j^{pred}$ **then**

8             $l_i^e \leftarrow l_i^e + l_j'^{pred}$

9             $b_i^e \leftarrow b_i^e + 1$

10        /*Check if over estimate output length*/

11        **if** $l_j^{real} < l_j^{pred}$ **then**

12            $l_i^e \leftarrow l_i^e + l_j^{real} - l_j^{pred}$

13            $b_i^e \leftarrow b_i^e - 1$

14  Calculate the equivalent error function $\alpha_i l_i^e + \beta_i b_i^e + c_1 = 0$ of worker $i, i = 1, 2, \ldots, I$. according to Eq. 4.

15  /*Fix error by adjusting the new requests placement*/

16  **if** *new request* $j$ *from worker* $x$ *to worker* $y$ **then**

17     $b_x^e \leftarrow b_x^e - 1$

18     $b_y^e \leftarrow b_y^e + 1$

19     $l_x^e \leftarrow l_x^e - l_j^{pred}$

20     $l_y^e \leftarrow l_y^e + l_j^{pred}$

21  /*Minimize the sum of the shortest distance between each worker's error function and the origin.*/

22  $min(\sum \frac{|c_i|}{\sqrt{\alpha_i^2 + \beta_i^2}}), i = 1, 2, \ldots, I.$

23  **Return** $x_{ij}, j = 1, 2, \ldots, J_{new}$

---

# F   System Design

Benefiting from the predictable nature of individual and batch LLM inference, we attempt to reveal the best way to serve requests that arrive as a stream from resource management and request placement perspectives. In this section, we describe the system design of Aladdin for two variances settings: default continuous batching and split-phase inference. The default continuous batching will process the input tokens and generate output tokens in the same worker, represented by vLLM Kwon et al. (2023). The split-phase inference refers to the inference setting that splits the prompt processing and token generation into different working instances, and each instance only processes prompt or generates output. This setting is represented by Splitwise Patel et al. (2023) and DistServe Zhong et al. (2024).

## F.1   System Workflow.

**Default continuous batching.** The Figure 5 illustrates the workflow of continuous batching inference scheduling. Firstly, users submit their LLM inference requests via the API as the first step ①. The request scheduler uses the bin packing heuristic to schedule the new requests according to their input length and the predicted output length ②. Lastly, the request scheduler continuously update the performance model according to the worker's execution traces ③.

**Split-phase inference.** Figure 6 illustrates the workflow of split-phase inference. Users submit requests through API ①. We schedule the prefill of new requests based on their input lengths. Since the prefill only involves one iteration, there is no queue for the prefill workers ②. Next, the decoding scheduler places the requests from prefill workers to decoding workers based on the pre-

Figure 5: Workflow of Aladdin with default continuous batching

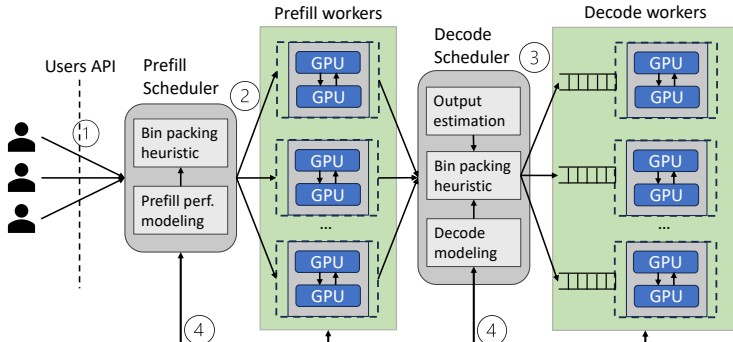

Figure 6: Workflow of Aladdin with split-phase inference.

dicted output length and a learned performance model ③. Finally, the prefill and decode schedulers continuously update the performance model according to the workers' execution traces ④.

### F.2 ADAPT TO CHANGING DEMAND

In every cluster heartbeat, we can reconfigure the cluster using change point detection. In LLM inference, although users submit different queries and receive different answers, the input and output lengths of LLM inference requests for the same model exhibit a strong pattern. From the SharGPT dataset Chiang et al. (2023), we found that the input lengths of user queries follow a fixed distribution, and the output lengths of the same LLM also follow a learnable distribution. According to our experiment using Algorithm 1, when the arrival rate $r_a$ is larger than a lower bound $R$, the total number of required workers $N_w$ is linear with the request arrival rate $r_a$.

$$N_w = \lceil k_5 r_a + c_5 \rceil, \; r_a > R \tag{7}$$

where $k_5$ and $c_5$ are learnable coefficients associated with the historical demand, and we round the number of workers to the smallest integer larger than the function of $r_a$. The reason $R$ exists is that when the arrival rate is lower, there are fewer requests arriving in the same heartbeat, which cannot represent the real distributions of the input and output length. The standard error of the mean $SEM = \frac{\sigma}{\sqrt{n}}$ is the metric for the difference between the sampled requests' input and output lengths and the total requests, where $\sigma$ is the standard deviation of all requests' input and output length and $n$ is the number of requests we place during a heartbeat. The smaller $n$ is, the more error appears in the prediction of $N_w$.

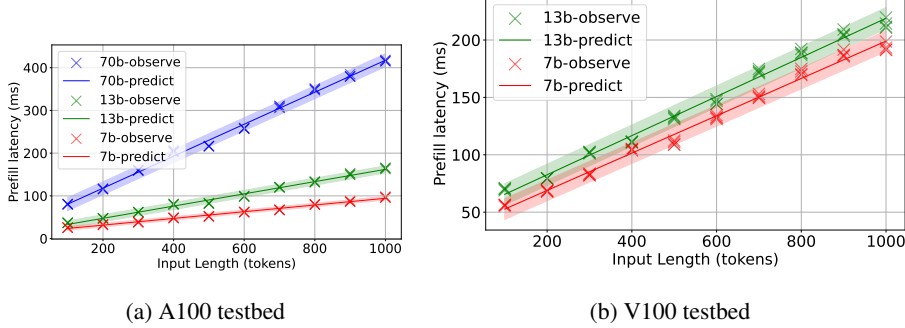

(a) A100 testbed

(b) V100 testbed

Figure 7: Prefill latency

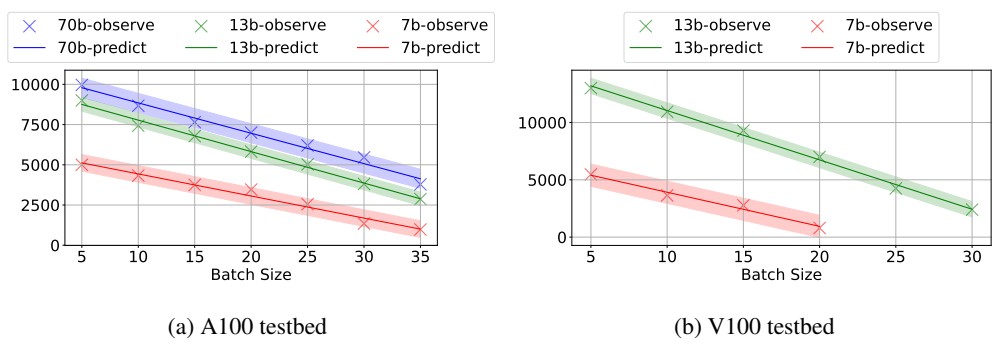

(a) A100 testbed

(b) V100 testbed

Figure 8: Decode context length limitation

With this model, we can predict the total number of workers required before placing all requests to each worker. However, the scheduling time requirement of inference serving is in milliseconds. In a high-demand situation, the scheduling overhead is too large to schedule the requests in the target iteration for the centralized scheduler. We design a distributed scheduler for the high-demand scenario that harnesses the pattern of input and output length of requests in Appendix L.

Note that in this paper, we focus on predicting the minimal GPU required for the varying arrival rate without considering the cold start problem and the switching cost. Since the optimization of cluster scheduling is orthogonal to the worker number prediction problem, we defer it to future work.

## G  TAIL LATENCY REMARKS

We consider a scheduling window on worker $j$ with $n$ co-scheduled requests. Let $\mathbf{a}_j \in \{0,1\}^n$ denote binary placement indicators (we also allow the relaxed $\mathbf{a}_j \in \mathbb{R}^n_+$ for analysis), and let $\mathbf{X} \in \mathbb{R}^n$ be the random vector of per-request end-to-end latencies (TTFT or ATGT) predicted by our performance model. Define $S_j := \mathbf{a}_j^\top \mathbf{X}$, $\mu_j := \mathbb{E}[S_j]$ and $\Sigma := \mathrm{Cov}(\mathbf{X})$ so that $\mathrm{Var}(S_j) = \mathbf{a}_j^\top \Sigma \mathbf{a}_j$. We allocate a global violation budget $\delta \in (0,1)$ across workers with nonnegative weights $\{\beta_j\}$ satisfying $\sum_j \beta_j \leq 1$, and set $\tau_j := \beta_j \delta$. Unless stated otherwise, we do not assume independence; covariances are estimated from traces or our stage-wise model. This appendix uses no notation that conflicts with the main text.

For any random variable $Y$ with mean $\mu$ and variance $\sigma^2 < \infty$, Cantelli's inequality gives

$$\Pr(Y - \mu \geq t) \leq \frac{\sigma^2}{\sigma^2 + t^2} \quad \text{for all } t > 0.$$

Setting $t = \sqrt{\frac{1-\tau}{\tau}}\,\sigma$ yields $\Pr\big(Y \geq \mu + \sqrt{\frac{1-\tau}{\tau}}\,\sigma\big) \leq \tau$. Applying this to $Y = S_j$ with $\sigma = \sqrt{\mathbf{a}_j^\top \Sigma \mathbf{a}_j}$ gives

$$\Pr\Big(S_j \geq \mu_j + \gamma(\tau_j)\sqrt{\mathbf{a}_j^\top \Sigma \mathbf{a}_j}\Big) \leq \tau_j, \quad \gamma(\tau) = \sqrt{\frac{1-\tau}{\tau}}.$$

To enforce a concrete SLO threshold $T_j$ (for TTFT or ATGT), we further require

$$\mu_j + \gamma(\tau_j)\sqrt{\mathbf{a}_j^\top \Sigma \, \mathbf{a}_j} \ \leq \ T_j,$$

which then implies $\Pr(S_j \geq T_j) \leq \tau_j$.

When a normal approximation is appropriate, we may instead take $\gamma(\tau) = z_{1-\tau} = \Phi^{-1}(1-\tau)$, the $(1-\tau)$-quantile of the standard normal, which is also a valid one-sided bound under Gaussian assumptions.

Let $\mathcal{E}_j := \{S_j \geq \mu_j + \gamma(\tau_j)\sqrt{\mathbf{a}_j^\top \Sigma \mathbf{a}_j}\}$. By the union bound (Boole's inequality),

$$\Pr\Big(\bigcup_j \mathcal{E}_j\Big) \ \leq \ \sum_j \Pr(\mathcal{E}_j) \ \leq \ \sum_j \tau_j \ = \ \sum_j \beta_j \delta \ \leq \ \delta.$$

Thus $\sum_j \beta_j \leq 1$ is a sufficient condition to keep the cluster-level violation probability within the global budget $\delta$.

Instantiating $S_j$ with TTFT or ATGT ties the chance constraints to concrete SLOs. Batch sizing and request placement modify $\mathbf{a}_j$ and hence both $\mu_j$ and $\mathbf{a}_j^\top \Sigma \mathbf{a}_j$, while worker configuration alters feasible $\mathbf{a}_j$ through compute and memory limits especially the KV capacity. Our heuristic rules (mixing long-prompt with long-output, capping decode iteration time, etc.) can be viewed as variance-reducing choices for $\mathbf{a}_j$, which tighten the tail bound without increasing mean load.

If per-request latencies are weakly correlated, $\Sigma$ is close to diagonal and

$$\mathbf{a}_j^\top \Sigma \, \mathbf{a}_j \approx \sum_{k=1}^n a_{jk}^2 \, \Sigma_{kk}.$$

If correlations exist like due to shared KV or interconnect contention, they are captured in $\Sigma$ and handled by the covariance-aware bound above. In practice we fit per-request means and $\Sigma$ from traces or our stage-wise model and apply shrinkage for robustness.

Under non-degenerate variability and calibrated $\gamma$, variance-aware mixing weakly improves tail feasibility for a fixed mean load compared with grouping alike requests. Consequently, for a target global budget $\delta$, the chance-constrained scheduler does not worsen the number of active workers nor the SLO-attainment rate relative to point-estimate baselines, and is often strictly better.

## H    EXPERIMENTAL SETUP

**Testbed setup.** We test the performance of Aladdin on high-end GPU servers with 4 A100 80GB GPUs connected with PCIe. Each machine has two Intel Xeon Platinum 8380 processors and 512GB RAM. To validate the generalization of Aladdin from both a computation perspective and communication perspective, we also evaluate Aladdin on the GPU servers with 4 V100 32GB GPUs connected with NVLink. Each machine has two Intel Xeon Gold 6230 processors and 128GB RAM. We also do a large-scale simulation for the high-demand request arrival situation.

**Models and SLOs.** Our target is to prove Aladdin reduces the cost for transformer-based LLMs. To validate that Aladdin can accurately model the performance metrics of most models. We evaluate Aladdin on Llama2 series Touvron et al. (2023) models from 7B to 70B. The model, testbed information, and SLOs are shown in Table 3. Note that the prefill latency SLOs are the approximated inference latency for the model's context length (4096 tokens) for each testbed. The selection of decode latency SLO is according to the individual request inference latency. We guarantee that the batch inference latency of each request won't exceed the individual inference latency for 1.3 times.

**Workload.** For the end-to-end performance evaluation in Section 5.1, we first collect the prompts

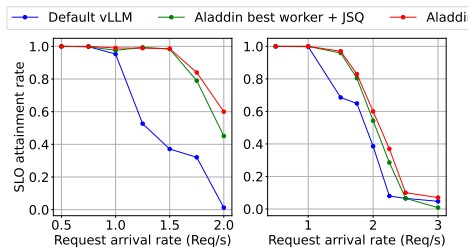 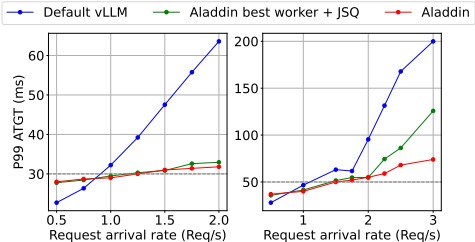

(a) The end-to-end SLO attainment rate, (left): LlaMa2-7b, (right): LlaMa2-13b

(b) The end-to-end P99 ATGT, (left): LlaMa2-7b, (right): LlaMa2-13b

Figure 9: End to end experiments on V100 testbed

from users of ShareGPT_V3_unfiltered_cleaned_split dataset teams (2023), then submit the prompts follows a Poisson distribution. The outputs are generated by each evaluated model with a temperature of $0$ and a maximum output token limit of $2048$. For the large-scale simulation in Section J, we use the same prompts' lengths as those collected from ShareGPT teams (2023) in Section 5.1 as the prompt lengths. Then, we predict the output length based on the output length CDF of the responses generated in Section 5.1's end-to-end evaluations for each model.

Table 3: The LLM information and testbed allocation

| Model | Testbed | Prefill SLO(ms) | Decode SLO(ms) |
|---|---|---|---|
| Llama2-chat 70b | A100 | 1600 | 75 |
| Llama2-chat 13b | A100, V100 | 600, 800 | 30, 50 |
| Llama2-chat 7b | A100, V100 | 400, 800 | 15, 30 |

Because there is no available trace of LLM inference that includes the arrival time of each request, we simulate the request arrival stream using a Poisson distribution. We need to validate that Aladdin improves performance in both high-demand and low-demand scenarios. To evaluate the performance of Aladdin with varying demands, we tune the average arrival rate $\lambda$ to simulate different request demands.

**Metrics.** Since our final target is to reduce the cost of the inference service, we use the number of GPUs required to achieve a certain SLO attainment rate as the main metric. In Section J, we evaluate the total GPU number required with different request arrival rates. In Section 5.1, As the total resources are limited for the real testbed evaluation, we evaluate the performance of Aladdin with the SLO attainment rate and the P99 ATGT in different request arrival rates. The SLO is attained when both TTFT and ATGT latency meet the requirement.

**Baselines.** Aladdin is a cluster-level scheduler. The performance improvement achieved by Aladdin is orthogonal to the improvements achieved by single-server optimization techniques such as split-phase inference Patel et al. (2023); Zhong et al. (2024) or page attention Kwon et al. (2023). These single-server optimization techniques use naive cluster scheduling like JSQ. Previous work Hu et al. (2024) adopts the power-of-two scheduling for request placement. However, it is suboptimal for request placement and cannot guarantee a high SLO attainment rate. We compared Aladdin's request placement with JSQ and power-of-two algorithms with different GPUs and different serving scenarios.

## I   PERFORMANCE MODEL VALIDATION

Request placement and worker configuration depend on accurate predictions of performance metrics. In this section, we evaluate the model's accuracy by comparing the predicted metrics to the measured actual metrics.

In Section 3, we model the latency of prefill phase and decode phase separately because the two phases have different characteristics. In the evaluation, we evaluate the prefill and decode latency

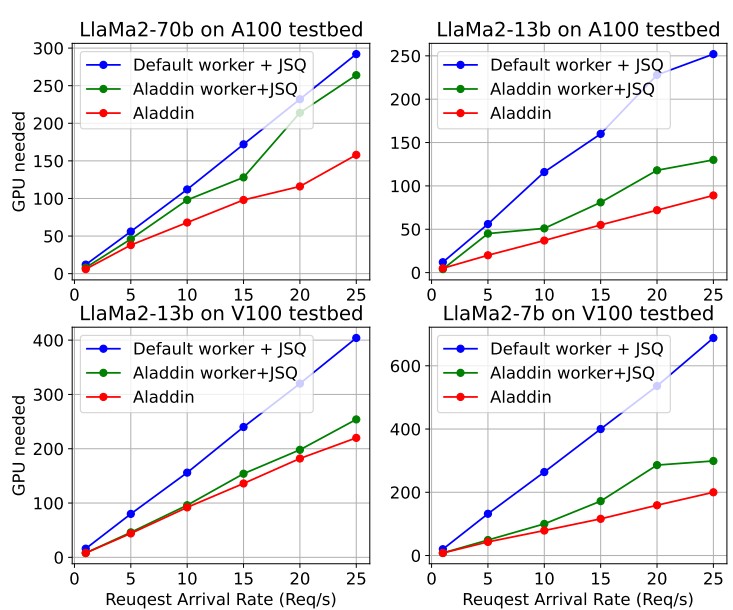

Figure 10: Simulation of the total GPU number needed with the mixed prefill and decode setting.

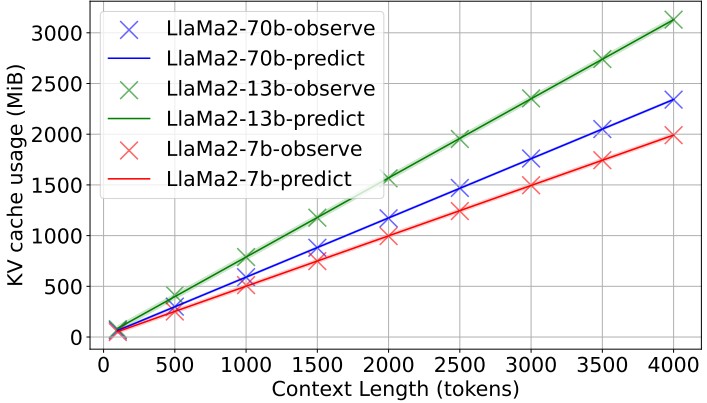

Figure 11: KV cache prediction and the observation.

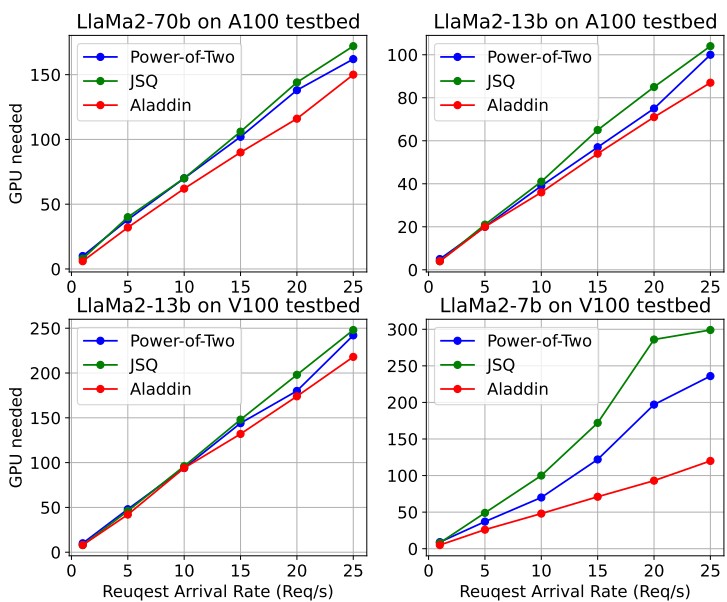

Figure 12: Simulation of the total GPU number needed for the decode phase of the split-phase inference setting

separately for different input and context lengths. In our prefill latency model, the prefill time of a batch size only corresponds to the total input length of all requests in the batch, not related to the batch size. In our experiment, we test different batch sizes $1, 2, 4, 8$ with the same total input length within a batch to validate this formulation. We only evaluated the LlaMa2-70b model on the A100 testbed because our V100 testbed could not load the 70b model (around $140GB$) even with all GPUs ($32GB*4$). Figure 7a and Figure 7b shows the results on A100 and V100 testbeds. The maximum prefill latency prediction error is less than $4\%$. The shaded area is the prediction interval, which represents the estimation of the range in which future observations are likely to fall. Results indicate the maximum error of the prediction interval compared with our prediction is less than $10$ tokens.In the decode latency model, the iteration time is linear with respect to both the batch size and the total context length within the batch, not related to the context length of each request in the batch. This means that regardless of whether the context length of each request in a batch is long or short, the decoding latency will be the same when the sum of the context lengths of all requests and the batch size is the same. In our experiment, for the same batch size, we use the same sum of context length but different context length distributions for all requests in a batch to validate this formulation. Results are presented in Figure 8a and Figure 8b. Similar to the prefill latency prediction, the prediction error is less than $5\%$. For the prediction interval, the error is less than 300 tokens for all context tokens in the batch.The KV cache usage to the context length is the most accurate metric in our performance models. According to Figure 11, the prediction error is less than $1\%$, and the prediction interval is just aligned with the prediction model. Note that the KV cache usage is not affected by the testbed; it is only related to the model. Generally speaking, the larger the model is, the more KV cache is needed for the same context length. However, from Figure 11, we can see that for the same context length, Llama2-13b requires more KV cache than Llama2-70b. This is because Llama2 7b and 13b adopt multi-head attention, while the 70b model adopts grouped-query attention Ainslie et al. (2023), which shares key and value pairs within each group.

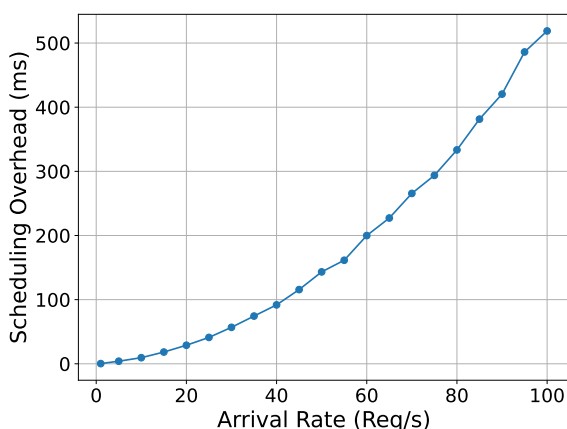

Figure 13: The bin packing algorithm running time with different arrival rates

## J   LARGE-SCALE SIMULATION

We conducted a simulation for the high-demand request arrival scenario. In this simulation, we evaluated Aladdin's performance with split-phase inference and the default vLLM inference setting. To show the direct cost savings of Aladdin, we simulate the GPU number required for P100 SLO-guaranteed inference serving at the different request arrival rates.

**Default Continuous Batching Inference.** In Figure 10, we compared vLLM with baselines in Section 5.1. Results indicate that Aladdin reduces the LLM serving cost by up to 71% and 40% compared with the default vLLM and JSQ with Aladdin optimal workers.

**Split-Phase Inference.** Previous work Patel et al. (2023); Zhong et al. (2024); Hu et al. (2024) split the prefill phase and decode phase into different instances. Split-phase serving maintains a group of prefill workers and a group of decode workers, as shown in Figure 6. According to the results of DistServe Zhong et al. (2024), the majority of GPU resources are scheduled for the decode workers. Since the scheduling for prefill instances is trivial with known prompt lengths, we only simulate the GPU number required for the decode phase instance. The baselines are JSQ adopted by DistServe Zhong et al. (2024) and the Power-of-Two algorithm adopted by previous work Hu et al. (2024). Results indicate that Aladdin reduces the GPU number required for the SLO-guaranteed decode phase by up to 60% and 49% compared with JSQ and Power-of-Two algorithm.

## K   SCHEDULING OVERHEAD

The scheduling overhead can be a problem in high-demand scenarios. For the scheduling latency, each scheduler's scheduling latency is predictable based on the request arrival rate since the time complexity of the best-fit bin packing algorithm is O(nlogn). Figure 13 shows the scheduling overhead in centralized scheduling. According to the results, with a request arrival rate of around 25 requests per second as we adopted in Section J. The scheduling overhead is less than 50 ms, which is acceptable. However, if the arrival rate is very high or the scheduling latency limit is very strict, we can follow Appendix L to adopt the distributed grouped scheduling.

## L   DISTRIBUTED SCHEDULING

The scheduling time requirement of inference serving is in milliseconds. In a high-demand situation, the scheduling overhead is too large to place the requests in the target iteration for the centralized scheduler. We design a distributed scheduler for this case that harnesses the pattern of input and output length of requests.

With a high arrival rate, the worker required for inference service is linear to the request arrival rate as discussed in Section F.2. Hence we can randomly sample the arrived requests into groups

using round robin. Then for each group, we only place the requests within the group to the workers corresponding to this group. While the arrival rate is $r_a$, if Group $i$ is corresponding to $N_i$ workers, the arrival rate of Group $i$ is $\frac{N_i}{N_w}r_a$. For this distributed scheduling mechanism, the scheduling demand of Group $i$ is $\frac{N_i}{N_w}$ of the total demand. Therefore, the scheduling latency is reduced.

The selection of group numbers is non-trivial. With more request groups, the scheduling overhead is low because each group has fewer requests. However, more workers are needed to fulfill the SLOs because the workers' utilization is reduced compared with centralized scheduling. Assume half of the groups require one extra worker to fulfill the SLOs according to the probability that the requests placed to these groups require more computing resources, to limit the resource error to less than $e$ in percentage, each group should be equipped with at least $\frac{1}{2e}$ workers. For example, if we want the extra worker required for serving compared with centralized scheduling is $10\%$, then each scheduling group should have at least five workers. For the scheduling latency, each scheduler's scheduling latency is predictable based on the request arrival rate since the time complexity of the best-fit bin packing algorithm is O(nlogn). To guarantee both the scheduling latency and extra resource error, the request arrival rate of each scheduling group $r_i$ is constrained by:

$$\frac{1}{2e} \le r_i \le r(T_s), \ i = 1, 2, \ldots, N_{group},$$
$$\sum r_i = r_a, \qquad i = 1, 2, \ldots, N_{group}, \tag{8}$$

where $N_{group}$ is the group number, $T_s$ is the scheduling latency limit. The $r(t)$ is the function of the arrival rate limit to the scheduling latency.

With the distributed scheduling, the end-to-end co-adaptive scheduling of Aladdin is described in Figure 14. The first step is to predict the optimal worker configuration according to Section 4.1 and the corresponding performance models based on Section 3. Given the request arrival rate, we predict the total worker number $N_w$ using Eq. 7 and search for the group number $N_{group}$ using Eq. 8. Then we use a round-robin router to route arriving requests to groups of schedulers. Finally, each scheduler packs requests to their corresponding workers using Algorithm 1.

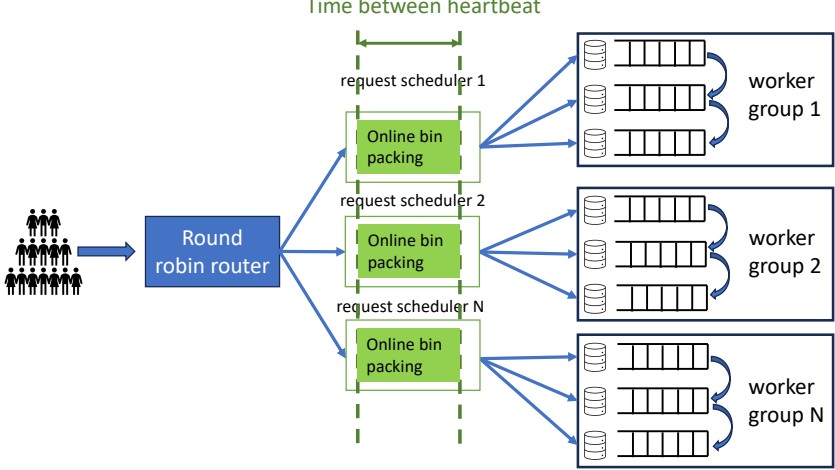

Figure 14: Distributed scheduling using request sampling.

## M ADDITIONAL EXPERIMENTS

In order to support the effectiveness of our framework, we add here some more experiments.

### M.1 DIFFERENT MODEL

The evaluation is not only limited to the Llama2 model family. Even though the input/output length distributions or performance characteristics, might be different for other modern architectures, the

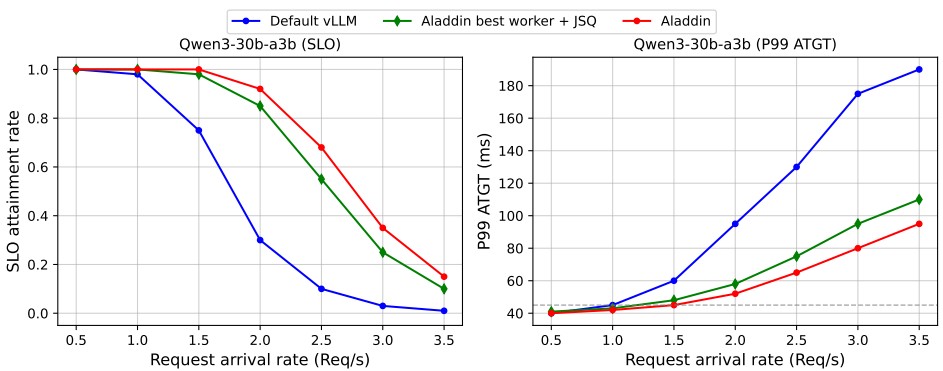

Figure 15: qwen3 test using A100

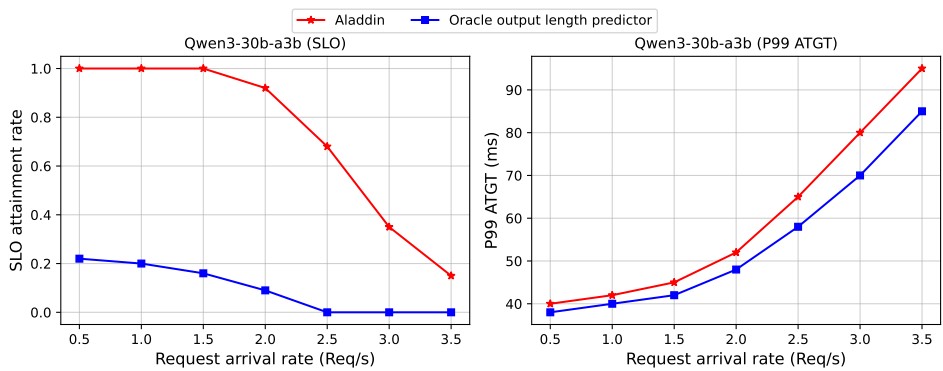

Figure 16: compare with oracle output length predictor

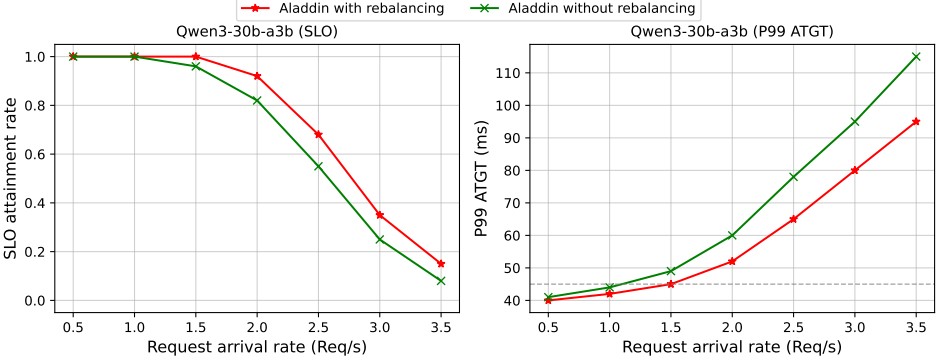

Figure 17: with and without rebalancing

modeling structure depends only on token linearity and KV scaling. To claim generality, we test Qwen3 Yang et al. (2025) on the same A100 testbed. From Figure 15 we can see that our algorithm achieves a similar advantage than default on both SLO attainment rate and P99 ATGT, just as in the Llama2 experiments. It validates that our proposed performance models and scheduling gains are not specific to Llama2.

## M.2 ORACLE PREDICTION

To clarify the role of prediction, we evaluate our algorithm with the output length oracle, in comparison to wtih our historical estimator. From Figure 16 we can see a lead of our algorithm over oracle, especially in SLO attainment rate. This is because the current length prediction oracle incurs a heavy latency.

## M.3 ABLATION OF REBALANCING

In the last example, we evaluate the necessity of our rebalancing mechanism. Figure 17 shows such a comparison, between Aladdin with and without rebalancing. The results shows that with rebalancing, Aladdin performs non-negligibly better in SLO attainment rate and ATGT than without rebalancing, over varying request arrival rate range.

