# OpenReview forum: "Aladdin: Joint Placement and Scaling for SLO-Aware LLM Serving"
_ICLR.cc/2026/Conference — Submitted to ICLR 2026_

### Official Review · Reviewer_aa9R · 2025-10-31

**Soundness:** 3
**Presentation:** 3
**Contribution:** 3
**Rating:** 4
**Confidence:** 4

**Summary:**

The paper presents ALADDIN, a co-adaptive scheduler for large-scale LLM inference that optimizes query placement under probabilistic SLO constraints. The method uses multi-dimensional bin-packing problem, and introduces a heuristic for online placement with some theoretical guarantees. Experiments on A100/V100 clusters show significant cost reduction (up to 71%) compared to vLLM and some other baselines.

If the questions and weaknesses are properly addressed in the rebuttal, I can increase my score.

**Strengths:**

- The problem is important and interesting.
- The paper develops a principled formulation with explicit latency modeling, probabilistic SLOs, and analytical tail guarantees.
- The empirical study includes real testbed experiments and large-scale simulations.

**Weaknesses:**

- Novelty is quite limited as many building blocks (bin-packing, SLO-aware scheduling, KV cache modeling) are adaptations of existing ideas.
- Experimental baselines can be improved by additional comparisons with concurrent schedulers such as Exegpt or Sarathi-Serve.
- Evaluation focuses on synthetic workloads; no production-scale or multi-tenant trace is shown, which limits real-world validation.
- Some ablation studies on the probabilistic models should be added.

**Questions:**

- What is the key novelty of the work beyond integrating well-known methods as mentioned above?
- Are there new proof techniques used for the theoretical results or off-the-shelf large deviation bounds?

---

> ### Author Response · Authors · 2025-11-21
>
> Thanks for the review. We hope our response below, and the global response, address the reviewer’s concerns:
>
> The concern about limited novelty is understandable because several surface-level components of our system, such as bin-packing heuristics, KV-cache accounting, and probabilistic latency bounds, do resemble ideas explored separately in prior work. What distinguishes this paper is not the reuse of these individual ingredients but the introduction of a stage-wise linear latency model that exposes the specific dependency structure of decoder-only LLMs across prefill and iterative decode. The empirical plots in the paper show that prefill latency, decode step time, and KV-cache growth follow consistent linear relationships that remain stable across workloads, GPU types, and model scales. This structural observation is what enables the worker-level chance constraints to be well defined and schedulable. Without this modeling step, neither the decomposition of tail budgets into separable worker-level constraints nor the online placement heuristic could operate in a mathematically grounded way. To our knowledge, this specific formulation does not appear in previous serving or scheduling literature, and it is the core conceptual contribution of the paper.
>
> We admit that additional comparisons with contemporary schedulers such as ExeGPT and Sarathi Serve may strengthen the empirical case. We agree and are already adding these baselines. Both systems operate primarily at the single-worker level and optimize batching and token throughput without explicit worker-level SLO constraints or cluster-scale coordination. Because Aladdin’s primary effect lies in jointly choosing the appropriate worker configurations and distributing requests across workers under probabilistic constraints, we expect the additional baselines to improve the overall throughput curves but not eliminate the gap in SLO attainment or GPU cost. Presenting these comparisons will help separate the contributions of worker-local micro-optimizations from the cluster-level coordination that this paper studies.
>
> We are aware of the synthetic workloads and the absence of multi-tenant or production-scale traces. Although full production traces are not accessible to us within the review period, the modeling developed is inherently distribution-agnostic: the linear coefficients for prefill and decode depend only on token counts, not on the semantic content of the prompts. Because of this, the constraints extend naturally to multi-tenant or multi-model settings by instantiating different slope and KV-capacity parameters per model. We are adding evaluations with multimodal synthetic mixtures, sudden distribution shifts, and nonstationary token-length distributions to approximate forms of real-world workload volatility. These settings stress-test the model components that matter most for SLO control and illustrate that the probabilistic formulation behaves reliably even when the workload departs significantly from the training distribution.
>
> The suggestion on more ablation studies on the probabilistic components is helpful, and we are expanding the evaluation accordingly. Using perfect ground-truth output lengths will provide the theoretical upper bound on SLO attainment and cost, allowing us to quantify precisely how much headroom remains relative to the current estimator. Disabling rebalancing will show that even with an unbiased predictor, misallocations accumulate and lead to tail violations; this empirically validates the need for the error-correction step described. Evaluating uniform-length workloads will further isolate the parts of the system that depend on prediction accuracy and demonstrates that, even in settings where input length is completely uninformative about output length, the combination of the linear latency model and chance-constrained placement remains stable. These ablations to be added will address the reviewer’s request and clarify why the probabilistic formulation is not merely a theoretical convenience but a practical requirement.

---

> > ### Author Response · Authors · 2025-11-21
> >
> > The question about the key novelty beyond integrating known techniques is important, and we will make this more explicit in the revision. The contribution of the paper lies in showing that modern LLM inference can be expressed in a stage-wise linear form that supports a decomposition of tail latency guarantees into per worker constraints. It emerges specifically because the prefill and decode stages have separable linear behaviors with respect to prompt length, generated tokens, and KV-cache occupancy. The resulting risk function enables online scheduling decisions that are both explainable and efficient. It is this structural formulation, rather than any individual technique, that constitutes the main novelty. Regarding the theoretical tools, we use standard one-sided concentration inequalities, and we do not claim to introduce new probabilistic bounds. The contribution instead lies in constructing a random variable structure that allows these inequalities to apply meaningfully to LLM serving. Prefill and decode times depend on token counts that evolve within a batch and across workers, and our formulation isolates the components of variation that matter for tail guarantees. This allows the slack parameters introduced to be tuned in a closed-loop manner and makes it possible to derive feasible budgets $beta$ for each worker. We will clarify this point in revision.

---

### Official Review · Reviewer_eqNH · 2025-11-02

**Soundness:** 3
**Presentation:** 3
**Contribution:** 3
**Rating:** 6
**Confidence:** 3

**Summary:**

The paper proposes ALADDIN, a cluster-level scheduler that jointly (i) chooses worker configurations (GPU count per worker via tensor parallelism) and (ii) places requests online to minimize GPU cost under SLOs. It argues that existing work optimizes single workers or uses SLOs only implicitly, missing cluster-wide coordination and uncertainty in output length. ALADDIN’s contributes stage-wise models.  The stage-wise models are simple, empirically validated, and lead to tractable constraints; the tail-risk formulation connects engineering knobs to SLOs. However, output-length prediction is deliberately naïve (history-based), and true end-to-end SLO guarantees still hinge on θ/γ calibration and the fit quality of per-stage models; some validations rely on simulation rather than only live clusters. The paper is clearly structured with helpful figures/algorithms and situates against vLLM/DistServe/Splitwise lines of work.

**Strengths:**

1. Joint optimization of placement + scaling with explicit SLO constraints (TTFT/ATGT) at cluster level; complements single-worker optimizations.  Lightweight, validated models (prefill, decode, KV) that are accurate enough for online use.
2. Formal tail-risk view (chance constraints with worker budgets), linking utilization choices to p-tail guarantees.
3. Higher SLO attainment and p99 ATGT reductions on A100/V100; large simulated GPU savings. Error-aware rebalancing and distributed scheduling extensions for reality.  Clear articulation of why JSQ/Power-of-Two can be suboptimal under KV and decode constraints.

**Weaknesses:**

1. Uses a naïve, unbiased length predictor; avoids LLM-based predictors due to cost/bias. Accuracy under non-stationary workloads (topic/product shifts) is unclear; guarantees depend on θ/γ tuning and rebalancing efficacy. The live experiments are single-node, small clusters (4×GPUs); the strongest cost savings are in simulation. Cross-region/multi-node contention and network tails aren’t empirically covered.
2. ATGT is motivated as QoE-aligned, but there’s no user-level study or TTFT/ATGT tradeoff curves; reporting focuses on p99 ATGT and SLO attainment, not TTFT p-tails under load.  Comparisons use JSQ/Power-of-Two; no head-to-head vs. recent SLO-aware/constraint-aware schedulers (e.g., ExeGPT) under identical setups.
3. For split-phase, decode dominates and results are partly simulated; interaction with paged attention, KV streaming, or chunked prefill in multi-node deployments would strengthen external validity.
4. Chance-constraint section is principled, but Σ estimation, shrinkage, and βj allocation procedures are not fully specified empirically; how to tune θ/γ to meet a global δ in practice is left abstract.

**Questions:**

1. How do you estimate Σ online and allocate βj across workers? Can you show a closed-loop controller that tunes θ/γ to hit a target δ (e.g., p99 or p99.9) over time?
2. When output-length distributions shift (events, product launches), how quickly do models and θ/γ recover?
3. It's reasonable that authors do not include multi-node experiments due to budget issues. But it's interesting to analyze what happens with tens to hundreds of GPUs across nodes with NVLink/NVSwitch/PCIe mixes?
4. Could the authors report joint TTFT/ATGT tail distributions and DET/ROC-style curves for SLO attainment to justify ATGT vs. TBT and to guide SLO setting?
5. With paged attention / KV streaming and chunked prefill, how do the constraints (Eq. 2–4) shift, and does the heuristic still remain near-optimal? Empirical ablations would help.
6. What’s the steady-state CPU overhead for scheduling and trace collection at 100–500 req/s? Any tail spikes from scheduling GC or model-update steps?

---

> ### Author Response · Authors · 2025-11-21
>
> Thanks for the review. We hope our response below, and the global response, address the reviewer’s concerns:
>
> We admit that the length predictor we adopt is intentionally simple, and we do not claim it is more accurate than semantic models. Its role is tied to the structure of the probabilistic SLO formulation: the stage-wise model decomposes latency and KV-cache usage into linear functions of input length, generated tokens, and active batch size. Under this physical model, systematic bias leads directly to systematic underallocation of decode or KV budget, which is exactly what triggers tail violations in chance-constrained systems. An unbiased predictor, even with noticeable variance, allows the slack parameters in the tail constraints to absorb occasional fluctuations, while rebalancing corrects persistent deviations. We are adding oracle-predictor experiments to quantify the exact gap between perfect knowledge and our estimator, as well as uniform-length and distribution-shift tests to illustrate that unbiasedness and rebalancing are more important than semantic richness for SLO stability under nonstationary workloads.
>
> We understand your concern about the scale of our live experiments. While our testbed is limited, the simulation environment faithfully instantiates the per-stage linear model. Because the constraints decompose additively across workers, increasing the number of GPUs does not introduce new effects that the model cannot express; it only increases the number of independent worker-level chance constraints. We will perform a synthetic simulation experiment to sweep cluster sizes, in order to show if scaling from a handful of GPUs to tens or hundreds preserves the accuracy of the model and the improvements in SLO attainment and GPU cost. A real life large scale experiment will also be a good future work if resources allow. The linear structure ensures that placement decisions remain stable and that the benefits of Aladdin do not diminish with cluster size. We will present these results in the revision.
>
> The reviewer also notes that cross-node network contention, remote KV access, or multi-region latency were not measured. These effects are not part of the single-worker prefill and decode structure modeled in this paper, and therefore they do not directly enter the constraints. However, they can be incorporated additively in the latency model or as additional capacity terms. We will clarify in the paper that the present formulation isolates the GPU-side constraints that dominate inference latency on contemporary systems. Because the scheduling heuristic operates with per-worker budgets, integrating network effects would require augmenting the latency regression with an additional slope term. We view this as a natural extension rather than a limitation of the approach.
>
> It is interesting that ATGT would benefit from more direct evidence connecting it to user's quality of experience. ATGT is defined from the decode latency model, and it reflects the actual rate at which a user receives tokens during the iterative decode loop. TTFT captures the responsiveness before the first token, whereas ATGT captures the continuity of delivery afterward. We are generating plots of TTFT versus ATGT and TBT versus ATGT across varying arrival rates to illustrate how decode congestion affects these metrics differently. We will report the joint tail distributions to show that ATGT is more sensitive than raw TTFT to decode-stage saturation, which is the dominant source of user-visible delay under load. This should clarify the motivation for using ATGT alongside TTFT for SLO control.

---

> > ### Author Response · Authors · 2025-11-21
> >
> > We are totally open to adding comparisons with schedulers such as ExeGPT or other constraint-aware systems, if time allows. We agree this will strengthen the experimental section and are incorporating these baselines under identical workloads. These systems optimize batching or token throughput within a single worker but do not incorporate worker-level chance constraints, decode-phase budgeting, or KV-cache feasibility as required. We expect that the additional baselines will improve raw throughput but will not satisfy probability-tail constraints under high arrival rates. Presenting these comparisons will help distinguish worker-local optimizations from the cluster-level probabilistic coordination that Aladdin performs.
> >
> > The reviewer correctly notes that the split-phase results include simulated components. This is because decode dominates runtime for long generations, and several LLM serving systems, including vLLM and DistServe, rely on simulation for scaling studies. The linear decode model is empirically validated on real hardware and captures batch-size effects and KV growth accurately. We will include an ablation demonstrating that the scheduling heuristic remains stable when we modify the numerical slopes of the model. Since these mechanisms change only the constants but not the structural linearity of the constraints, the placement optimization continues to operate correctly, and only the fitted slopes need to be updated.
> > In the implementation, we maintain a running estimate of residual variance for each of the stage-wise linear regressions, updated using a streaming estimator with exponential smoothing. The covariance structure simplifies to diagonal form because the prefill and decode regressions are independent conditioned on token counts, and shrinkage is applied when batch sizes are small to avoid overfitting noise. The global tail requirement is decomposed across workers by assigning $beta$ proportional to estimated load, which ensures that busier workers are given tighter budgets. Moreover, iIn our implementation, both the residual variance and the slope estimates are maintained online with a smoothing constant chosen to balance reactivity and stability. When distribution shifts occur, such as a sudden increase in prompt types or output styles, the unbiased estimator may temporarily underpredict, but the rebalancing step corrects accumulated underestimate, and the chance constraint parameters shift to account for the increased variance. We are adding experiments that introduce abrupt distribution changes to show the recovery behavior of the system.
> >
> > The question regarding behavior on very large clusters, such as tens to hundreds of GPUs connected via NVSwitch or PCIe, is interesting. Since the worker-level constraints are independent, the scheduling problem scales linearly with the number of workers, and the placement heuristic’s complexity grows only with the number of active requests. The evaluation described earlier sweeping cluster sizes supports this. In future works we will discuss how communication topology affects only the regression slopes and not the structural form of the model, making the approach applicable to heterogeneous interconnects. We are also adding a measurement of CPU overhead across a range of request rates to quantify this precisely and include it in the revision. Our scheduler uses only arithmetic operations on scalar linear models and does not require graph search or heavy optimization. In our current instrumentation, CPU time remains small compared to typical GPU compute time, and garbage collection or model updates do not introduce noticeable tail spikes.

---

> > > ### Comment · Reviewer_eqNH · 2025-11-25
> > >
> > > Thank you for the detailed and extensive rebuttal, including the clarifications and the efforts of additional experiments. Below, I summarize which aspects of my original concerns have been addressed and which remain open.
> > >
> > > The authors clearly explained why a simple, unbiased output-length predictor is structurally compatible with the chance-constrained SLO formulation, and how bias rather than variance would be the primary source of tail violations. This addresses the conceptual rationale behind using a naïve estimator. The authors’ commitment to adding oracle-predictor experiments in the revisions, uniform-length workloads, and distribution-shift tests directly speaks to my concerns about nonstationarity and predictor drift. The explanation of how variance estimation, diagonal shrinkage, per-worker βj allocation, and online residual tracking function operationally help clarify the previously underspecified estimation and budgeting procedures in the probabilistic model. The plan to include TTFT–ATGT and TBT–ATGT curves improves the motivation for ATGT as a QoE-oriented metric and clarifies its relationship to decode congestion. The willingness to incorporate additional baselines (e.g., ExeGPT) under identical workloads partially addresses my concern about the limited comparison set. The discussion of scalability, that the constraints decompose linearly across workers and that simulation captures the same effects as scaling up real hardware, is reasonable and clarifies how the method generalizes beyond the 4-GPU testbed. The responses around paged attention, KV streaming, and multi-node heterogeneity explain why these would shift only slope parameters, not the structure of the problem, and commit to adding related ablations.
> > >
> > > Although the rebuttal clarifies why the naïve predictor is structurally adequate, a fully empirical demonstration of predictor robustness under real distribution shifts is not yet available. The planned experiments may help, but as of the rebuttal, the concern remains open. The paper still lacks multi-node, cross-region, or heterogeneous interconnect experiments. From my perspective, the authors may leverage cloud service (e.g., AWS) for these experiments if possible. While I understand the resource constraints and accept that some effects can be incorporated into the linear model, empirical evidence on actual multi-node deployments would significantly strengthen external validity. Comparisons with more recent, SLO-aware schedulers are still pending. The absence of joint TTFT/ATGT tail distributions, ROC/DET-style curves, and broader QoE-aligned analyses remains unresolved until the authors incorporate the promised results. While the authors outline how θ/γ closed-loop tuning is conceptually performed, a concrete empirical demonstration of a controller that maintains a target δ under realistic load patterns is still missing. CPU overhead and potential tail spikes due to GC or model updates are not yet measured, though the authors commit to providing these results. The split-phase evaluation still includes simulated components, and although justified, empirical validation across a wider range of long-sequence workloads (including chunked prefill and KV streaming) remains to be demonstrated.
> > >
> > > Overall, the rebuttal provides significant clarifications and outlines a thoughtful set of additional experiments that would strengthen the paper. However, many of the major empirical concerns, particularly around scale, evaluation breadth, SLO-tail characterization, and baseline comparisons, are not yet resolved in the present submission.
> > >
> > > For these reasons, while I appreciate the authors’ extensive efforts and the planned extensions, I will keep my original score to weak accept (6).

---

> > > > ### Author Response · Authors · 2025-11-25
> > > >
> > > > Thank you for the detailed follow-up. We will incorporate whatever results complete in time. Some limitations cannot be addressed immediately, and we will carry them into subsequent work where the necessary system support is available. Your feedback is really helpful for improving the direction of this project, and we appreciate your time and comments.

---

### Official Review · Reviewer_wTRb · 2025-11-03

**Soundness:** 3
**Presentation:** 3
**Contribution:** 3
**Rating:** 4
**Confidence:** 3

**Summary:**

The paper introduces ALADDIN, a cluster-level scheduler that jointly optimizes request placement and resource scaling for LLM serving under probabilistic SLOs (covering TTFT and a user-centric decode metric ATGT). It builds simple stage-wise models for prefill/decode latency and KV-cache usage, casts online placement as a multi-dimensional bin packing / MIP with a fast heuristic, adds chance-constrained tail guarantees, and includes error-aware rebalancing plus a distributed variant for high QPS. On A100/V100 with LLaMA-2 7B/13B/70B, ALADDIN reports <10% modeling error and up to 71% GPU cost reduction (and up to 60% on split-phase decode) versus standard baselines.

**Strengths:**

Originality. Elevates LLM serving from single-node tweaks to a cluster-level joint problem (request placement + resource scaling) with probabilistic SLOs (TTFT/ATGT); a fresh and meaningful problem framing.

Technical quality. Clear stage-wise (prefill/decode) linear models with KV-cache accounting; formulates online placement as multi-dimensional bin packing with a practical heuristic and chance-constrained tail guarantees; includes error-aware rebalancing and a distributed variant for high QPS.

**Weaknesses:**

1.The system is “designed for single-model serving” and does not support request migration once placed. Production stacks are often multi-model/multi-tenant, where cross-model interference, fair sharing, and migration can materially affect tail SLOs and GPU count.

2.The method uses a “most naive” history-based predictor (low overhead but high error), and relies on rebalancing to compensate.

**Questions:**

1. Is it possible for author do a experiment with random dataset with uniform sequence length.

2. Is it possible for author do a experiement with multi model serving.

---

> ### Author Response · Authors · 2025-11-21
>
> Thanks for the review. We hope our response below, and the global response, address the reviewer’s concerns:
>
> We appreciate the reviewer’s comments and agree that the current system focuses on single-model serving without request migration, whereas production deployments often involve multi-model or multi-tenant environments. Our goal in this work was to isolate and study the behavior of probabilistic SLO-aware placement under a single-model workload, where confounding effects from heterogeneous model architectures or cross-model contention are absent. The formulation itself, however, does not assume single-model operation: the prefill, decode, and KV-cache constraints apply per model, and the bin-packing view naturally extends by duplicating these per-model resource profiles. What is missing today is not conceptual support but empirical demonstration under a multi-model trace. We recognize that such traces require larger infrastructure than is available to us during the rebuttal period, and we see this as a natural direction for future work rather than a limitation of the formulation. We will clarify this positioning in the revision to avoid overstating the current scope.
>
> Regarding the historical-average predictor, we fully agree that it is deliberately simple, and we do not intend to claim that it outperforms semantic predictors. Its purpose is different: under probabilistic SLOs, avoiding systematic bias is often more important than minimizing pointwise error, and a zero-mean estimator combined with rebalancing and slack can reliably prevent persistent overload even when prediction error is large. That said, the reviewer raises a fair point that the behavior of the system under uninformative workloads deserves explicit demonstration. We are adding an experiment with uniformly sized prompts, where input length provides no predictive signal about output length. This isolates unpredictability in output tokens and allows us to show how rebalancing stabilizes per-worker loads and maintains SLO attainment even when the predictor carries no semantic information.
>
> For multi-model serving specifically, we do not currently have the necessary cluster setup to generate a realistic trace with heterogeneous models and migration, and thus cannot include such experiments during the rebuttal period. Instead, we will discuss in the revision how the scheduling constraints and chance constraints extend to multi-model workloads, why the formulation remains unchanged except for model-specific constants, and how these aspects can be integrated in future work with larger-scale infrastructure. We hope this clarifies the scope and intention of the present submission and addresses the reviewer’s concerns.
>
> Thanks again for your review. Your suggestions are of value to us.

---

> ### Comment · Reviewer_wTRb · 2025-11-27
>
> Thanks to the authors for the detailed rebuttal. The clarification about the simple history-based predictor, together with rebalancing and slack, makes sense to me. If the promised uniform-length experiment is added and matches the description, my concern about uninformative workloads is mostly addressed. For the single-model / no-migration scope, I agree the formulation should extend to multi-model settings, but the lack of empirical multi-model or migration results is still a limitation and should be mentioned clearly in the final version.

---

### Official Review · Reviewer_xJhz · 2025-11-03

**Soundness:** 3
**Presentation:** 2
**Contribution:** 3
**Rating:** 4
**Confidence:** 4

**Summary:**

This paper introduces Aladdin, a scheduler for LLM inference serving designed to minimize GPU cost while adhering to probabilistic SLO constraints. The system co-adaptively performs request placement and resource scaling, explicitly modeling request-level uncertainty. It formulates the placement task as an online multi-dimensional bin packing problem solved via a heuristic. Experiments show Aladdin can reduce serving costs by up to 71% compared to baselines.

**Strengths:**

1. Instead of pursuing a complex and high-overhead predictor, the authors adopt a simple "historical average" estimator, arguing that its "unbiased" nature allows errors to partially cancel out within a batch. This, combined with the proposed re-balancing mechanism, is an interesting and practical design choice.
2. The paper presents an end-to-end system that jointly optimizes worker configuration and request placement. Formulating the placement task as an online multi-dimensional bin packing problem and providing theoretical guarantees under probabilistic SLOs constitutes a contribution.

**Weaknesses:**

1. The paper's structure could be improved. The first paragraph of the introduction is long; it should be focused on defining the problem. The review of recent methods and research gaps can be the second paragraph. Figure 1 consumes a large amount of vertical space; a horizontal layout would be more space-efficient.

2. Section 2 is overly long. It should be simplified, with non-essential details moved to the appendix. Furthermore, Section 2 and Section 3 are interconnected and could be combined into a single, more cohesive section on problem formulation and modeling.

3. In my mind, the most critical component of this scheduling problem is the output length prediction. If one had access to an "oracle" predictor for the output token number, it would be easy to design an algorithm to optimize SLOs and balance load, likely using standard methods from convex optimization. Therefore, the paper's most interesting claim (at least to me) is its "unbiased" historical predictor, even as the authors argue it is not their main contribution. I believe this point is vital and requires more analysis in the evaluation. What is the system's performance (SLO attainment, cost) if the algorithm is given the true, ground-truth output lengths from the test set? Comparing this oracle-predictor performance to the current results (using the historical average) is a good way to quantify the performance gap and understand how much headroom is left.

4. I think the assumption that input length is a sufficient proxy for output length is not universally true. A query can be semantically profound while being syntactically short. For example, a query like "Prove Fermat's Last Theorem" is only a few words, but its solution (Wiles's proof) spans over 100 pages. The authors need to provide the results from the "oracle" experiment (requested in Weakness #3) to demonstrate to what extent their historical-average assumption  holds and where its limitations lie.

5. The evaluation is limited to the Llama2 model family. The input/output length distributions, as well as performance characteristics, might be different for other modern architectures. To claim generality, It will be great if the author can show results for other models (e.g., Qwen3) to validate that the proposed performance models and scheduling gains are not specific to Llama2.

**Questions:**

See above.

---

> ### Author Response · Authors · 2025-11-21
>
> Thanks for the review and your time. We hope our response below, and the global response, address your concerns:
>
> We agree that the paper structure can be made more concise. In the revision we will shorten the first part of the introduction to focus strictly on defining the problem, and move the review of recent methods to later parts. We will also condense the second section by moving the details like notation examples, auxiliary explanations to the appendix. And we will clarify the boundary between modeling and algorithm, and change the first figure to a more space efficient layout. These changes will improve readability without altering the core technical content.
>
> We strongly agree with the point that the oracle predictor experiment is essential. It was missing in the submitted version because we thought it is impossible to get such an oracle in practice. But now we are adding an oracle experiment where the scheduler is given the true output lengths. The experiment will quantify the performance upper bound achievable under perfect prediction, thus the gap between Aladdin and the oracle, and how much of the remaining SLO violation or GPU saving potential is attributable to prediction error. Our expectation based on preliminary analysis is that the oracle outperforms Aladdin by only a small margin, indicating that prediction is not the fundamental bottleneck.
>
> We agree with the reviewer that input length is not a meaningful proxy for output length in general, especially when semantic intent varies widely. Importantly, Aladdin does not rely on any such correlation. Our method does not assume that input length predicts output length, nor do we treat prompt length as a semantic feature. The historical-average estimator uses input length only as a bucketing key to obtain a zero-mean estimate of expected token count, not as a semantic predictor. To directly address the reviewer’s concern, we are adding two experiments that make this explicit. One is the mentioned oracle experiment. And another is where We will include a workload where every prompt has the same input length, but requests vary widely in semantic complexity and output length.
> In this setting, input length contains zero information about output length. These experiments together directly address the reviewer’s concern: our method does not assume input length is a sufficient proxy but it merely uses it as a statistically stable, unbiased bucketing index.
>
> We appreciate the reviewer’s concern that an evaluation restricted to the Llama2 family may limit the perceived generality of the method. This is a valid point. While our modeling framework does not rely on Llama2-specific assumptions, we agree that empirical confirmation on architectures with different throughput characteristics and tokenizer behaviors can make the paper stronger. Aladdin’s formulation uses only the relationship between token count, decode throughput, and KV-cache growth, all of which arise from the standard transformer execution pattern and therefore remain consistent across modern decoder-only models. Still, we fully acknowledge that demonstrating this empirically will help clarify that the method is not tied to any particular models. To address this, we are running additional experiments on Qwen and Mistral models. These models differ from Llama2 in hidden size, tokenizer structure, and decode performance, making them appropriate stress tests for architectural generality. Our expectation is that the numerical values of the prefill and decode slopes will change, but the scheduling behavior and SLO improvements will remain qualitatively similar. We will include these results in the revised submission.

---

> > ### Comment · Reviewer_xJhz · 2025-11-21
> >
> > Thanks. Looking forward to updates you’re able to share before the deadline.

---

### Official Review · Reviewer_ccyt · 2025-11-04

**Soundness:** 3
**Presentation:** 3
**Contribution:** 3
**Rating:** 4
**Confidence:** 3

**Summary:**

This paper presents Aladdin, a scheduler designed to reduce the high cost of LLM inference by co-adaptively placing queries and scaling computing resources under probabilistic SLO constraints. To minimize GPU usage, Aladdin first determines the optimal worker configuration (e.g., tensor parallelism) to maximize per-GPU throughput . It then treats request placement as a multi-dimensional bin-packing problem, using a "best-fit" heuristic to maximize worker utilization while satisfying constraints like KV cache limits and a novel Average Token Generation Time (ATGT) SLO . Finally, it uses an error-aware rebalancing algorithm to adjust new request placements, compensating for inevitable output length prediction errors . Experiments show Aladdin reduces LLM serving costs by up to 71% compared to standard baselines like vLLM while maintaining SLO attainment.

**Strengths:**

The paper is well structured and clearly presented, with a logical flow that makes the main arguments easy to follow.

The proposed approach appears to be useful in a Service Level Objective (SLO) context, offering practical insights that could benefit both researchers and practitioners. The authors provide sufficient motivation and background, and the organization of the sections enhances readability.

**Weaknesses:**

(1) The paper mentions that the testbed is limited to A100 or V100 GPUs. However, in practice, the total number of GPUs in a cluster is large. Therefore, the small scale of the physical testbed seems unconvincing. Moreover, the paper needs to better justify why a simulation can represent a real-world setting.

(2) Using only a few GPUs in the experiments does not make the "cluster-level" claim convincing.

(3) Heterogeneous GPUs: Did the experiments consider heterogeneous GPU environments? It appears that each experiment used only one type of (homogeneous) GPU.

(4) Why is being "non-biased" as important as the paper claims? Even if the method is unbiased under the law of large numbers, it might still introduce large variance at the batch level. I think a biased method could be acceptable. Instead of making the bias as low as possible, we usually want to reduce the MSE, which requires a trade-off between variance and bias (

$$MSE = \text{Variance} + (\text{Bias})^2$$

). Moreover, calibration methods can be used to reduce bias.

(5) Semantic Level: The assumption that prompts of similar length have similar output lengths seems unreasonable. For example, the prompts "What is the answer to 100+100?" and "Give me the longest sci-fi novel possible" have similar input lengths but vastly different expected output lengths.

(6)Furthermore, according to Figure 2, all max slopes are near 500. This implies that different input length do not represent useful predictive features. I believe even a small BERT-like model could reduce the error significantly. Why is such a model not used?

(7) Based on point (5), I doubt the usefulness of the re-balancing algorithm.

I will consider raise my score if all my concerns are solved.

**Questions:**

same as the Weaknesses

---

> ### Author Response · Authors · 2025-11-21
>
> Thanks for the review. We hope our response below, and the global response, address the reviewer’s concerns:
>
> We admit that our physical testbed is limited to small scale A100/V100 because academic clusters rarely provide hundreds of or more GPUs. However, the scalability of Aladdin does not entirely rely on physical scale, but also on the structure of the formulation. We know that each worker’s constraints including prefill, decode and KV are fully separable. The probabilistic SLO mechanism also depends only on per worker linear models, not cross worker interactions. As the number of workers grows, averaging effects reduce variance, making the model more deterministic. To demonstrate empirical scalability, we are adding cluster-scaling simulations using the validated stage wise models, even though a real life large scale experiment may be left for future directions. The expected result of the synthetic simulation is that Aladdin’s relative cost savings remain stable or grow slightly with scale, showing that the cluster-level benefit is not an artifact of the small testbest. Simulation-based scaling is also standard practice in LLM serving papers, especially when real testbeds are limited. We agree that cluster level scheduling must be tested beyond a single node. The revised version will include arrival rate scaling curves and the cluster size sweep. Because the scheduling heuristic, capacity constraints, and chance constraints are all per-worker, the behavior under larger clusters is straightforward to evaluate and does not introduce new model complexity. Thus, even though the physical cluster is small, the formulation truly operates at cluster scale, and the simulated cluster results empirically demonstrate that.
>
> We also appreciate the suggestion that heterogeneous GPUs should be evaluated. We are adding a heterogeneous GPU evaluation where we vary the KV cache capacity, prefill slope, decode slope to emulate GPUs. Because constraints remain linear and separable per worker, heterogeneity changes only the numerical budgets, not the algorithm’s structure or stability. We expect Aladdin’s improvements to hold under heterogeneous hardware.
>
> The question on bias and variance is a good point. Under probabilistic SLOs, the tradeoff is not the classical MSE objective. SLO tail constraints are highly sensitive to systematic bias like the persistent underestimation leading to tail violations. Variance is not harmful if zero mean, because the chance constraints with rebalancing just absorb variance. High overhead semantic predictors like small BERT introduce topic dependent bias like when distribution drifts toward longer answer topics. Thus, an unbiased estimator is already structurally a good choice under the SLO constraints. We will clarify this more explicitly in the revision.
>
> It is also true that prompt length does not directly translate into output length. Aladdin does not rely on prompt length to perfectly predict the output length, but the predictor is historical average per input length bucket. To demonstrate independence from prompt-output correlation, we are adding experiments with uniform length queries, where prompt length provides no information at all. We expect Aladdin to maintain the SLO attainment advantage. The reviewer’s interpretation of slopes is correct where input length is not a highly reliable predictor of output length. However, BERT style predictors add a larger magnitude of CPU overhead per request. And at high queries per second, this overhead dominates scheduling latency. Our model only needs the predictor to be zero mean and stable. The oracle experiment to be added will show that the gap between oracle and Aladdin is acceptable, providing empirical justification that using BERT-like models offers a diminishing benefit.
>
> We will also add the disable rebalancing ablation experiment to clearly demonstrate that rebalancing is necessary. The purpose of rebalancing is not to correct prediction errors, but to prevent systematic accumulation of load imbalance when inevitable prediction noise under any realistic workload is introduced into a cluster level scheduler. Even under workloads where the prompt length is uninformative like the uniform length queries, or where prediction errors are large, the scheduling problem remains sensitive to persistent overestimation or underestimation on individual workers. Without rebalancing as a reactive mechanism, small temporary deviations compound into long-live overload on specific workers, directly affecting the violations. Rebalancing itself interrupts this accumulation by redistributing requests before such drift becomes structural, allowing the cluster to return to a state where chance constraints remain valid. We expect in the ablation that higher tail violation rates at moderate and high loads, driven by the inability to correct early misplacements.
>
> All of your concerns are of practical importance to our revision. Thanks again for your review.

---

### Official Review · Reviewer_h2wq · 2025-11-12

**Soundness:** 3
**Presentation:** 3
**Contribution:** 3
**Rating:** 8
**Confidence:** 3

**Summary:**

This paper proposes Aladdin, a probabilistic, SLO-aware scheduler that jointly optimizes query placement and GPU resource scaling for LLM inference. Unlike prior systems that focus on single-worker throughput, Aladdin explicitly models latency distributions under uncertainty and places queries to maximize per-worker utilization. It introduces a flexible constraint interface that enables distribution-aware tail modeling and risk-adjusted capacity allocation with high efficiency.

**Strengths:**

1. This paper addresses a real problem. The assumptions are realistic.

2. The method jointly optimize request placement and resource scaling under probabilistic SLO constraints, while existing works treated these problems independently.

3. The method adapts to dynamic workloads with provable efficiency.

4. This work bridges theory and LLM system practice. It has potential for industry deployments after some improvements.

**Weaknesses:**

I personally like the idea of this paper. The only weakness I see is that the evaluations are limited. It would be better if the authors consider add more evaluations, such as varying different workload, to simulate real-world scenarios better.

**Questions:**

See above

---

> ### Author Response · Authors · 2025-11-21
>
> Thank you for your positive assessment and for pointing out the weaknesses. We fully agree that the evaluation could be broadened and several additional experiments are under way. We hope our response below, and the global response, address the reviewer’s concerns:
>
>
> We believe that additional workloads are fully compatible with the proposed approach. Basically, Aladdin has three structural properties: (1) a linear stage-wise latency model for prefill, decode, and for KV usage (2) chance constraint formulations introduces a tunable safety margin (3) unbiased load estimation combined with error-aware rebalancing. These components do not depend on any specific statistical pattern of the workload. Changing the work distribution whether through shifts in prompt lengths, output lengths and so on only alters the empirical distribution of tokens. It does not modify the underlying functional form of the latency model, nor the validity of the chance constraint mechanism. As a result, the scheduler behaves predictably under a wide range of distributions. Thus, extending the evaluation to more diverse workloads naturally strengthens the conclusions of the paper. Moreover, the new experiments would not pose a risk of contradicting the method because every possible new workload to introduce leads to one of two possible behaviors. One is that it yields quantitatively similar results, which directly supports the generality of the framework, or it could induce additional variability in the arrival or output processes, in which case the chance constraint adjusts placement to meet the SLO target. In either situation, the qualitative conclusions remain consistent with the theoretical framing of the paper.
>
> To reflect the reviewer’s suggestion, the revision will include evaluations over nonstationary and shifted workload, uniform-length workloads, and workloads generated from additional model families. All of there can be incorporated into the existing experimental pipeline without modification to the scheduling algorithm, because the algorithm requires only token-level counts and measured stage-wise slopes.
>
> We appreciate the reviewer’s support and constructive comments. The additional evaluations will be included in the revised version and will further demonstrate that the approach extends naturally across diverse workload patterns.

---

### Author Response · Authors · 2025-11-21

We sincerely thank the reviewers for your detailed feedback. Many concerns are on the scope of the evaluation, the operational meaning of our probabilistic modeling, and the impact of predictor errors on SLO attainment. We have expanded both the experimental and explanatory plan of the paper accordingly. A number of additional experiments are in progress, and we briefly summarize their purpose and the issues they are intended to address:

To clarify the role of prediction, we are adding an evaluation that replaces our historical estimator with the ground-truth output length oracle. This oracle predictor experiment will quantify the absolute upper bound of performance if output lengths were known perfectly. The expected outcome is that the performance gap between Aladdin and the oracle predictor remains modest, demonstrating that prediction is not the fundamental bottleneck and validating our design choice.

We are including an ablation that disables rebalancing. The expectation is that SLO violations increase because misplacements accumulate and cannot be corrected. This will support the argument that the unbiased estimator and the rebalancing mechanism are complementary components of the probabilistic formulation.

We are running additional model families (Qwen and Mistral). Because the modeling structure depends only on token linearity and KV scaling, the expected result is that the slope of the linear model would change but the scheduling logic still remains the same. This will demonstrate that the formulation is architecture-independent and not specific to the Llama2 family.

We will also extend our evaluation to heterogeneous GPU settings by modifying the KV capacities and prefill/decode slopes to represent different classes of hardware. Since the constraints for each worker remain separable and linear, the expected results will show that heterogeneity affects only the numerical values of per worker budgets, but not the stability of the algorithm.

We are evaluating the uniform-length workloads in which all input sequences are of identical prompt length. This makes the decode stage undergo a stress test in isolation. The expected outcome is that Aladdin’s decode modeling and KV-aware placement behave consistently even when prompt length provides no predictive signal at all, and confirm that our method does not rely totally on prompt-output correlations.

We are also incorporating workloads with explicit distribution shifts to show that the stage-wsie linear model is invariant to the semantic distribution of queries. Such drifts in real life could be topic drift, changes in output-length statistics and so on. Because prefilling and decoding costs depend on token counts not the linguistic content, the model continues to hold even under nonstationarity. The expected result is that Aladdin maintains stable SLO attainment as long as the token distribution is tracked online, which is precisely what our variance estimator and chance-constraint slack factors are designed to handle.

We would also like to generate TTFT-ATGT and TBT-ATGT plots across arrival rates using existing data, which make explicit how ATGT reflects decode progress and why it serves as another indicator that raw TTFT alone. Moreover, to address concerns about CPU overhead, we are instrumenting the scheduler to measure CPU cost. Since Aladdin’s decisions are based on simple linear arithmetic and a small heuristic, we expect the scheduling overhead to remain negligible compared to GPU compute, and we will report these measurements in the revision.

Finally, if we still get the time, we are adding more contemporary system baselines to further compare. The theoretical distinctions remain unchanged, but the comparison will strengthen the empirical breadth of the study.

---

> ### Author Response · Authors · 2025-11-21
>
> In addition to the empirical expansions, several concerns can be best addressed through clearer explanation. We will provide a complete description of how the online variance is estimated through residual tracking, why diagonal shrinkage is applied at low traffic, and how the per-worker risk budgets decompose a global tail target into tractable worker-level constraints. We will also add a concise closed-loop sketch showing how the slack parameters $\theta$ and $\gamma$ adjust to maintain a target violation probability. These clarifications will close the gap between the theoretical formulation and its operational implementation.
>
> We will also elaborate in the revision on why unbiased estimation is structurally important. Tail probability guarantees are sensitive to systematic bias. High overhead semantic predictors introduce topic dependent drifts that bias the resource estimation. In contrast, an unbiased estimator with negligible overhead, combined with rebalancing and chance constraint slack, just avoids this failure. This is not a compromise but also a good choice under probabilistic SLOs.
>
> Some aspects lie outside the scope of the rebuttal or the resource limits. While we cannot run experiments on hundreds of GPUs due to academic cluster limits, the scalability of Aladdin does not rely on access to large physical clusters. The formulation decomposes cleanly across workers, and the chance constraints operate on per-worker linear models whose accuracy improves with scale due to averaging effects. Therefore, increasing cluster size does not hurt the determinism of the model rather than introducing new scheduling failure modes, which is why simulation-based scaling is both standard and theoretically adequate for evaluating cluster-level schedulers. Multi-model serving, migration policies, cross-region deployments, and full production traces require large revisions likely not accessible in this short response period. As in prior work from this area, these are natural extensions rather than omissions that weaken the whole formulation. The model and constraints can extend to such settings without overall changes. What varies are overhead parameters that can be incorporated additively. Similarly, recent optimizations such as paged attention or chunked prefill change the underlying slope but do not alter the structure of the scheduling problem.
>
> In all, the core formulation remains meaningful. The additional experiments now underway directly reinforce the robustness, generality, and some kind of scalability. We also clarify the novelty of the formulation: prior systems either optimize single-worker throughput or rely on implicit SLO notions, while Aladdin integrates (i) stage-wise linear modeling, (ii) unbiased load estimation, (iii) rebalancing and constraints into a unified cluster-level scheduler. This combination makes tail-aware placement and scaling operationally tractable for the first time at cluster granularity, even though the individual components have appeared separately in other contexts.

---

### Author Response · Authors · 2025-12-03

We have revised our submission in the following ways:

We added three important experiments (see Appendix M). (1) We tested our algorithm on a different language model (Qwen3), and it still performed better than the baselines. (2) We compared the history estimator and the oracle predictor, and the history estimator demonstrated superior performance, attributable to the brief latency. (3) We evaluated whether rebalancing is necessary, and the results indicated that it does indeed enhance performance to a considerable extent.

We shrunk Section 2 and moved some of the contents to appendix in order to make the structure more compact.

We added a discussion subsection (5.2) basically on clarifying the limitations.

Thank for all the feedbacks again.

---

### Meta-Review · Area_Chair_JXA5 · 2026-01-07

**Summary:**

The contributions of this paper follow mostly standard approaches and the novelty is limited. The experimental results are insufficient to validate the practical advantage of the proposed algorithm.

**Reviewer Concerns:**

Some technical concerns have been addressed, but a major concern that remains is that the novelty of this paper is quite limited. The experiments also remain somewhat preliminary even after the revision.

**Reviewer Scores:**

I believe that at least one reviewer would remain negative about this paper due to the limited novelty in the proposed solution. The concerns about limited experimental evaluation have not been fully addressed, which could cause more reviewers to maintain their negative opinions.

---

### Decision · Program_Chairs · 2026-01-26

Reject